# Single-cell analysis reveals region-heterogeneous responses in rhesus monkey spinal cord with complete injury

Yongheng Fan[1,2,4], Xianming Wu[1,4], Sufang Han[3,4], Qi Zhang[1], Zheng Sun[1,2], Bing Chen[1], Xiaoyu Xue[1], Haipeng Zhang[1,2], Zhenni Chen[1,2], Man Yin[1,2], Zhifeng Xiao ®[1] ✉, Yannan Zhao ®[1] ✉ & Jianwu Dai ®[1] ✉

Spinal cord injury (SCI) leads to severe sensory and motor dysfunction below the lesion. However, the cellular dynamic responses and heterogeneity across different regions below the lesion remain to be elusive. Here, we used single-cell transcriptomics to investigate the region-related cellular responses in female rhesus monkeys with complete thoracic SCI from acute to chronic phases. We found that distal lumbar tissue cells were severely impacted, leading to degenerative microenvironments characterized by disease-associated microglia and oligodendrocytes activation alongside increased inhibitory interneurons proportion following SCI. By implanting scaffold into the injury sites, we could improve the injury microenvironment through glial cells and fibroblast regulation while remodeling spared lumbar tissues via reduced inhibitory neurons proportion and improved phagocytosis and myelination. Our findings offer crucial pathological insights into the spared distal tissues and proximal tissues after SCI, emphasizing the importance of scaffold-based treatment approaches targeting heterogeneous microenvironments.

Spinal cord injury (SCI) is a severe condition affecting the central nervous system (CNS), often resulting in permanent motor and sensory dysfunction[1,2]. Following SCI, blood-borne immune cells and stromal cells infiltrate the spinal cord and interact with glial cells, leading to the formation of dense scars[3,4]. Typically, mature SCI lesions can be categorized into three compartments: (1) a non-neural injured area located at the lesion core, also known as a fibrotic scar, consisting of various stromal and immune cells; (2) a dense glial scar separating neural tissue from the non-neural injured area, primarily composed of astrocytes and other glial cells; (3) spared but activated tissues surrounding the glial scar, comprising all cell types found in the normal spinal cord[5–7]. The pathological features of the tissues in these regions are heterogeneous.

Numerous studies have suggested that a series of cellular and molecular events occur in the spared tissues following SCI[8–10]. In recent years, several treatments targeting segments below the lesion have shown promise in improving locomotor function in animal models and patients with severe SCI, indicating that targeting the spared segments below the lesion could be a potential strategy for further repair of SCI[11–13]. Furthermore, functional scaffold implantation at the injury sites has demonstrated improved motor functions in rats, dogs, and humans with severe SCI[14–16]. However, minimal research attention has been given to the cellular heterogeneity and responses of different segments below the lesions, and the effects of treatment on the distal lumbar regions remain unclear.

[1]State Key Laboratory of Molecular Developmental Biology, Institute of Genetics and Developmental Biology, Chinese Academy of Sciences, 100101 Beijing, China. [2]University of Chinese Academy of Sciences, 100049 Beijing, China. [3]College of Animal Science, South China Agricultural University, 510642 Guangzhou, China. [4]These authors contributed equally: Yongheng Fan, Xianming Wu, Sufang Han. ✉e-mail: zfxiao@genetics.ac.cn; ynzhao@genetics.ac.cn; jwdai@genetics.ac.cn

Single-cell or single-nucleus RNA sequencing (scRNA-seq/snRNA-seq) has provided unprecedented insight into cell heterogeneity and dynamic processes in complex tissues. They have been extensively used to investigate the developmental and disease processes and cellular heterogeneity in the spinal cords of rodents and humans[17–24]. Following rodent SCI, single-cell molecular profiling has revealed temporal and spatial pathological alterations and cellular interactions in the injury area[25–28]. Recently, Matson et al. have profiled changes in cells within the lumbar spinal cord of mice after a thoracic injury and identified rare spinal neurons expressing a signature of regeneration in response to injury[8].

Given the anatomical and functional differences that affect the pathological process between rodents and primates, nonhuman primates are ideal animal models for studying SCI in the human spinal cord[29–31]. In this study, we utilized scRNA-seq/snRNA-seq to generate a single-cell atlas including regions proximal and distal from the lesion of the rhesus monkey spinal cord at 7 days, 14 days, 30 days, and 6 months after complete SCI. Our findings revealed that the spared lumbar tissue distal from the lesion was also affected, resulting in the formation of a degenerative microenvironment. Microglia and oligodendrocytes were induced into disease-associated phenotypes around descending neural fibers, while the percentage of inhibitory neurons in the distal lumbar tissues increased. White matter and gray matter astrocytes showed heterogeneous activation in the proximal tissues. Additionally, we found that functional collagen scaffold implantation in the injury sites could remodel the microenvironment of the injury sites and the lumbar tissues, highlighting the potential of scaffold-based treatment strategies for repairing SCI. Our study provides a valuable resource for dissecting cellular heterogeneity and microenvironments in different regions below the lesions of the primate spinal cord.

## Results

### Single-cell atlas of the injured spinal cord in rhesus monkeys revealed region-specific cellular dynamics

To clearly define areas after SCI, we created a complete injury by cutting and removing ~8 mm of thoracic (T9) spinal cord tissue in adult rhesus monkeys, resulting in a gap area named the injured area (IA). Immunostaining of the spinal cord tissues demonstrated sharp cell death (TUNEL staining), myelin debris, and activation of GFAP$^+$ astrocyte and AIF$^+$ microglia around lesions (Fig. 1a, b). The area 5 mm from the IA zone that contained massive myelin debris was named the degenerative area (DA), while the adjacent 5-mm-long area with obvious astrocyte activation was named the spared but activated area (SA). Additionally, we analyzed the spared lumbar (SL) tissues located ~8 cm away from the lesion (Fig. 1c).

snRNA-seq has the advantage of unbiased analysis of neural cells in the central nervous system[32,33]. However, we found that it was unsuitable for analyzing IA and DA regions in a complete transection SCI model because the extracted nuclei from these areas were mainly derived from apoptotic cells, which did not meet the requirements for library construction and sequencing. Therefore, we conducted snRNA-seq on SA and SL and scRNA-seq on IA and DA separately at the acute phase (7 days and 14 days) and chronic phase (30 days and 6 months). Additionally, we analyzed different regions (cervical, thoracic, and lumbar) of an uninjured spinal cord (0 days) with snRNA-seq for subsequent comparative analysis (Fig. 1c).

After excluding low-quality cells and potential doublets, we utilized 199,870 cells/nuclei for downstream analysis. We performed unsupervised clustering and annotated the cells by examining the expression of canonical marker genes. Our analysis included neurons, microglia/macrophages, oligodendrocytes, oligodendrocyte precursor cells (OPCs), astrocytes, ependymal cells, fibroblasts, pericytes, endothelial cells, Schwann cells, and T cells. Differential expressed genes (DEGs) analysis revealed unique molecular signatures for each cell type (Fig. 1d, e).

We observed obvious differences in the cellular constitution among various regions (Fig. 1f). The IA was primarily composed of fibroblasts, microglia or macrophages, vascular cells (including endothelial cells and pericytes), and T cells. At 14 days post-injury (dpi), T cells peaked at 32.5% and numerous Schwann cells migrated into the IA. Endothelial cells increased to 25.2% in the IA at 6 months. At 30 dpi, microglia increased from 25.2% at 0 dpi to 48.8% in the SA, and from 18.8% to 45.1% in the SL. In contrast, astrocytes decreased from 9.3% at 0 dpi to <2.4% and neurons decreased from 29.1% to 15.7% in SL (Fig. 1g).

To confirm that these results accurately reflected changes in tissue, we conducted immunostaining for neurons (NeuN), oligodendrocytes (APC), astrocytes (SOX9), microglia (AIF1), and fibroblasts (DCN) on IA, SA, and SL tissue sections (Supplementary Fig. 1a–d). Quantitative analysis showed similar trends to the snRNA-seq data; that is, microglia proportion increased in each region while other cells decreased correspondingly. However, we found that the density of astrocytes and neurons did not obviously change, indicating that the reduction in their proportion was not due to cell death, but rather the increase in total cell number caused by microglia.

### Injury led to the continuous activation of microglia into a disease-associated state

To understand the heterogeneity of microglia, we conducted a further cluster analysis of microglia/macrophage clusters and identified six distinct subtypes (Fig. 2a, b). Based on gene expression and gene ontology (GO) analysis of DEGs, we labeled clusters 1–5 as homeostatic microglia, inflammatory microglia, monocytes/macrophages, dividing microglia (DM), and interferon-activated microglia, respectively (Fig. 2c and Supplementary Fig. 2a, b). We also found that homeostatic microglia exhibited changes in gene expression after injury. For instance, almost all microglia did not express *CDH13*, even in the distal lumbar tissue at 6 months after SCI (Supplementary Fig. 2c, d).

Interestingly, we observed that the proportion of DM in SL (9.6%) was comparable to that in SA (10.4%) at 7dpi (Fig. 2d). Immunostaining results demonstrated that DM were predominantly located in the corticospinal tract (CST) area on lateral sides of the lumbar spinal cord (33.4%), rather than in the fasciculus gracilis (FG) area on the dorsal column (4.2%; Supplementary Fig. 2e).

We observed that cluster 0 downregulated homeostatic microglia genes and highly expressed *GPNMB* and *CTSD* (Fig. 2c), which are similar to disease-associated microglia (DAM) involved phagocytosis and lipid metabolism in neurodegenerative diseases[34–36]. Cluster 0 exhibited gene expression patterns resembling DAM (Fig. 2e), and the upregulated genes were enriched in GO terms such as phagocytosis, lipid transport, and lipid catabolic process (Fig. 2f). We named cluster 0 DAM-like microglia (DAMLM). The proportion of DAMLM among microglia in SL reached about 48%, which was equivalent to that in SA at 6 months (Fig. 2d). The number of changed genes in DAMLM compared to homeostatic microglia in SL was similar to that in SA during the early phase, but it had more altered gene in SL than SA during the chronic phase (Supplementary Fig. 2f). DAMLM expressed higher levels of genes involved in lipid transport in SA during the acute phase. In contrast, DAMLM in SL showed increased expression of phagocytic-related genes, which was further intensified during the chronic phase (Fig. 2g and Supplementary Fig. 2g, h). Interestingly, compared to SA, DAMLM in SL specifically upregulated genes related to responses to nutrient levels at both 7 dpi and 30 dpi (Supplementary Fig. 2g, h). This suggested that microglia in SL were in a phagocytosis state and had delayed but sustained responses to injury.

To investigate the location of DAMLM, we performed immunostaining of GPNMB. The results showed many spherical microglia clustered within areas of descending tract at 6 months, including the lateral and anterior CST area. However, little DAMLM was observed in

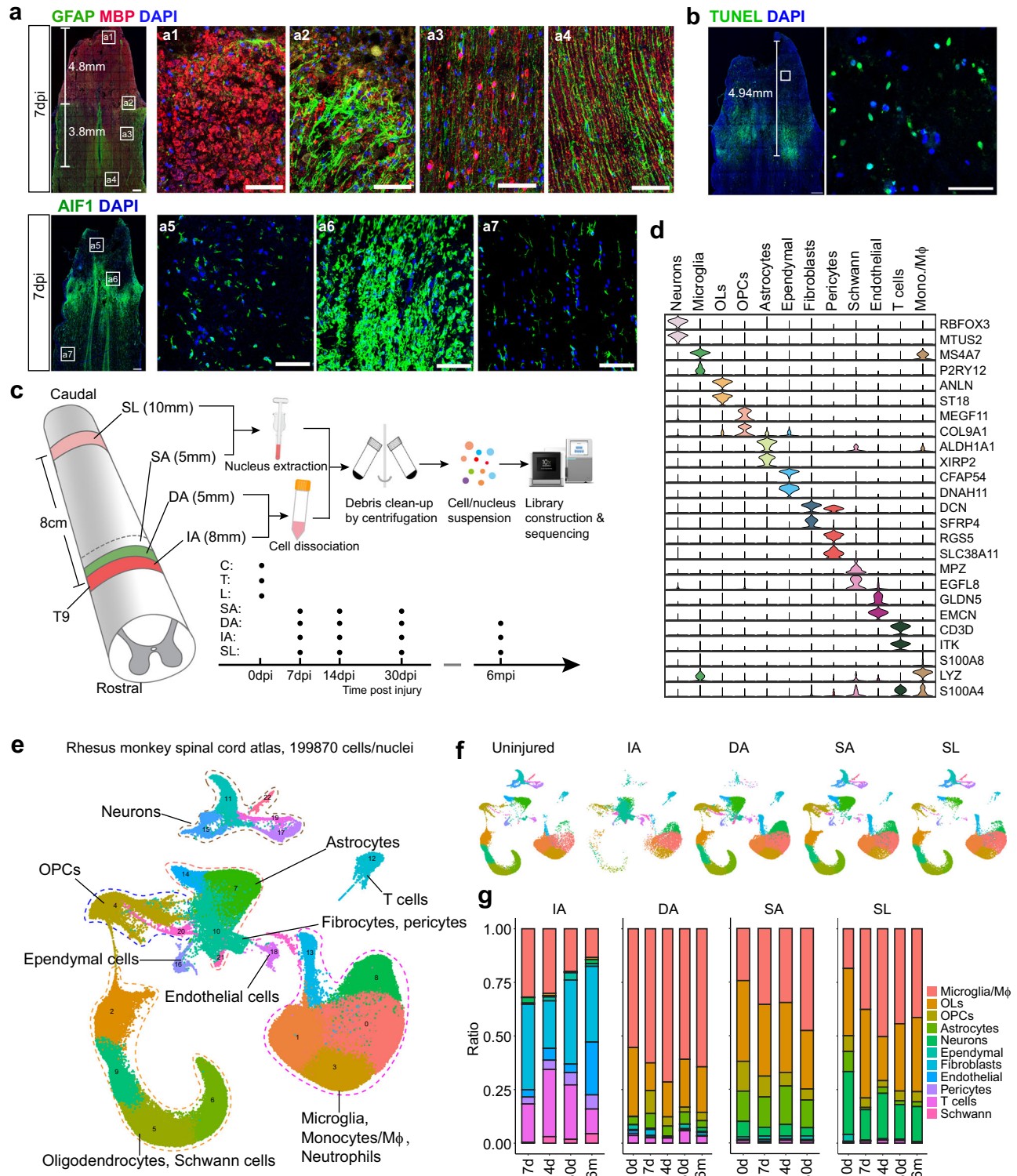

**Fig. 1 | Single-cell atlas of injured rhesus monkey spinal cord reveals region-related cellular dynamics from proximal to distal tissues. a** Immunostaining of GFAP, MBP, and AIF1 characterizing the different responses from proximal to distal spinal cord tissues in the injured rhesus monkey at 7 days. The magnified images highlight massive myelin debris region (a1), high expression of GFAP and AIF region (a2, a6), the activated MBP+ signals located in the cell body of oligodendrocytes region (a3), and the spared distal regions (a4). Scale bars: 500 μm in the left panel; 100 μm in a1-a7. **b** TUNEL assay indicating the apoptotic cells in the proximal injured rhesus monkey spinal cord at 7 dpi. The scale labels the regions of TUNEL-positive cells. The areas in white boxes are shown at high magnification. Scale bars, 500 μm in the left panel; 100 μm in the right panel. **c** Schematic overview of the

experiment design for single-cell sequencing of monkey spinal cord samples. Uninjured cervical (C), thoracic (T), and lumbar (L) spinal cord tissue and different regions (IA injured area, DA degenerative area, SA spared area, SL spared lumbar) of the injured spinal cord were harvested separately for single cell/nucleus sequencing at 7, 14, 30 days and 6 months post SCI. **d** Violin plots showing signature genes of cells in the rhesus monkey spinal cord. OLs oligodendrocytes, mono./Mϕ monocytes/macrophages. **e** Single-cell/nucleus atlas (UMAP plot) with 199870 high-quality cell/nucleus transcriptional profiles of intact and injured rhesus monkey spinal cord. **f** Split UMAP plots showing the cellular constitution of different regions in the injured rhesus monkey spinal cord. **g** Cell percentage dynamic of different regions in the injured rhesus monkey spinal cord.

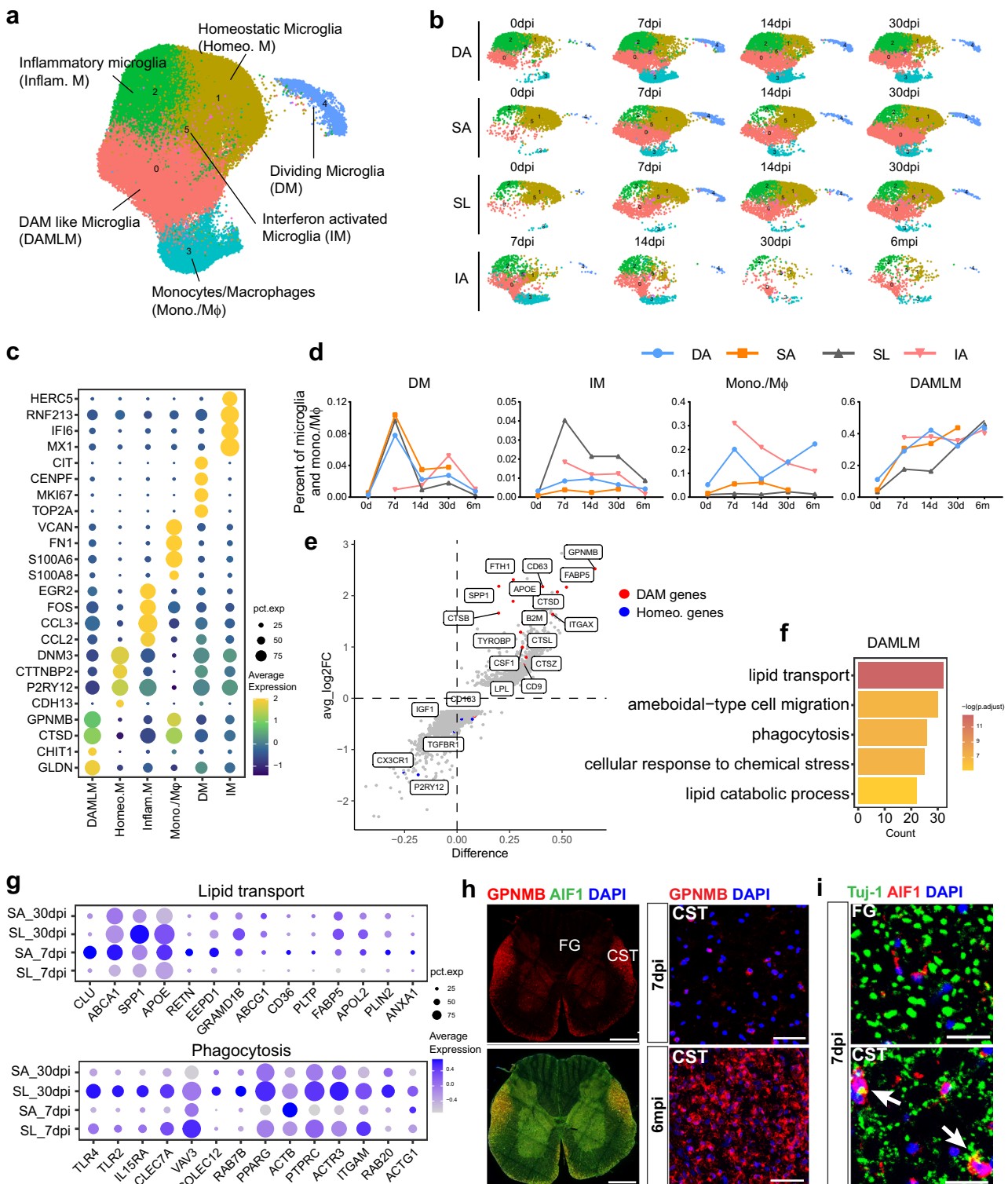

the areas of the ascending tract, such as the FG area (Fig. 2h). At 7 days post-injury, few GPNMB+ cells were present in the CST area, but they increased obviously at 6 months. We also examined the expression of Tuj-1 (labeled neurofilament) and AIF (labeled microglia) in lumbar sections at 7 dpi. The results showed that neurofilaments in the FG area remained intact, while irregularly shaped Tuj-1 positive signals appeared in the CST area, and some Tuj-1+ signals co-localized with microglia (Fig. 2i). This suggests microglia in the distal lumbar tissue were induced into a phagocytic phenotype surrounding descending axons after SCI.

## DAMLM and phagocytic macrophages exhibited low levels of pro-inflammatory genes

Following SCI, monocytes, and macrophages infiltrate the spinal cord parenchyma. We conducted a further cluster analysis of monocytes/macrophages and identified four subclusters, including neutrophils (*FRY*, *CSF3R*), monocytes (*FN1*, *PECAM1*), and two types of macrophages (*GPNMB* and *CCL3*) (Supplementary Fig. 3a–c). Interestingly, these two types of macrophages exhibited gene expression patterns resembling DAMLM and inflammatory microglia. We renamed them phagocytic macrophages and inflammatory macrophages,

**Fig. 2 | SCI massively activates microglia into DAM-like microglia (DAMLM) characterized by lipid metabolism and phagocytosis. a** UMAP plot depicting the heterogeneous clusters of microglia, monocytes, and macrophages in the intact and injured rhesus monkey spinal cord. DAM, disease-associated microglia. **b** Split UMAP plots showing the regional distribution of cell subsets at distinct time points after SCI. **c** Differentially expressed genes among microglia and macrophage subsets. The size of the dot indicates the percentage of cells in which that gene is detected and the color bar is scaled with the average expression of the corresponding genes. **d** Dynamic changes in the proportion of microglia subsets among microglia in different regions after injury. **e** Differential gene expression analysis between DAMLM and homeostatic microglia. Red dots and blue dots indicate the signature genes of DAMLM and homeostatic microglia, respectively. **f** GO enrichment analysis of the highly expressed genes in DAMLM compared with homeostatic microglia. *P* values (adjusted) were calculated using Benjamini–Hochberg false discovery rate (FDR). **g** Dot plots showing dynamic expression of genes for lipid transport and phagocytosis enriched in corresponding GO terms in DAMLM from SA and SL at 7 dpi and 30 dpi. The size of the dot indicates the percentage of cells in which that gene is detected and the color bar is scaled with the average expression of the corresponding genes. **h** Immunostaining of GPNMB (DAMLM marker) and AIF1 (microglia marker) in rhesus monkey lumbar spinal cord at 6 months post injury (left). The areas of CST are shown at high magnification (right). FG fasciculus gracilis, CST corticospinal tract, Scale bars: 1 mm in the left panels and 50 μm in the right panels. **i** Immunostaining of AIF1 and Tuj-1 in the FG and CST areas of the rhesus monkey lumbar spinal cord at 7 days after SCI. Scale bars, 50 μm. Source data are provided as a Source Data file.

respectively. Phagocytic macrophages were mainly distributed in DA and SA, whereas inflammatory macrophages primarily existed in the IA (Supplementary Fig. 3d). Similarly, phagocytic macrophages expressed many genes involved in lipid metabolic processes, while inflammatory macrophages expressed genes related to cytokine production and regulation (Supplementary Fig. 3e, f).

Microglia and macrophages play crucial roles in inflammatory regulation and can be polarized into a pro-inflammatory M1 or anti-inflammatory M2 phenotype[37,38]. We examined the expression of M1 and M2 marker genes and noted that inflammatory microglia expressed high levels of M1 marker genes (*IL1B*, *IL1A*, and *TNF*). However, DAMLM and homeostatic microglia exhibited low expression of M1 markers and high expression of M2 markers (*ARG1*, *MRC1*, and *CDH1*) (Supplementary Fig. 3g). Similarly, phagocytic macrophages expressed fewer M1 genes, while inflammatory macrophages expressed numerous genes of both M1 and M2 (Supplementary Fig. 3h). This suggests that DAMLM and phagocytic macrophages have similar functions with low expression of M1 genes.

## Heterogeneous responses of proximal and distal neurons after SCI

To better understand the heterogeneity of neurons in rhesus monkeys, we conducted cluster analysis and identified 28 distinct neuronal subclusters (Fig. 3a). Many neurons in the DA were not captured, consistent with the immunostaining results that few neurons survived in the DA area. Within a 3 mm region away from the IA, almost all neurons died by 7 days after SCI (Fig. 3b–d).

We conducted an analysis of neurons in the uninjured monkey spinal cord. No obvious differences were found in neuronal subsets among different segments (Supplementary Fig. 4a, b). We identified representative genes for each neuronal subtype (Supplementary Fig. 4c) and performed GO analysis of the top DEGs of all clusters, which suggested that neuronal diversity was primarily driven by genes encoding neurotransmitter receptors, ion channels, and transmembrane transporters in the membrane (Supplementary Fig. 4d). The cluster patterns of neurons were related to the neurotransmitter status and spatial location (Supplementary Fig. 4e-h). We renamed these neuronal subsets according to neurotransmitter status (E, excitatory; I, inhibitory; C, cholinergic; M, mixed) and spatial location (D, dorsal; M/V, middle/ventral) (Supplementary Fig. 4a, c). Immunostaining verified the distribution of the DI-5 subcluster (NPY⁺), which showed a similar distribution to mice on the superficial surface of the dorsal horn (Supplementary Fig. 4i). Hierarchical clustering also revealed that neuronal subsets sharing common neurotransmitters and locations were more closely related (Supplementary Fig. 4j). These findings were consistent with previous studies in mice[17,19,39], indicating the conserved neuronal architecture pattern in the spinal cord across species.

The cluster pattern related to the neurotransmitter status of neuronal subsets remained unchanged after SCI (Fig. 3a and Supplementary Fig. 5a). However, we observed that the proportion of excitatory and inhibitory interneurons changed differently proximal and distal to the lesion. In the SA, the proportion of excitatory interneurons increased (from 52.04% to 61.94%) while the inhibitory interneurons decreased (from 44.08% to 35.64%) at 7 dpi. This phenomenon was partially recovered at 30 dpi. In contrast, the proportion of both types of interneurons in SL did not change at 7 dpi. However, from 7 days to 6 months, the proportion of inhibitory interneurons increased (from 38.59% to 52.63%), while the excitatory interneurons decreased (from 54.44% to 44.74%; Fig. 3e and Supplementary Fig. 5b). Immunostaining with γ-aminobutyric acid (GABA), one type of inhibitory neurotransmitters, in the spared lumbar sections confirmed the increased proportion of inhibitory interneurons at 6 months after SCI (Fig. 3f). This suggests that change in neurotransmitter expression in the distal lumbar below the lesion were opposite to those in regions near the lesion.

Neurons in the SA had more altered genes at 7 dpi than those at 14 dpi and 30 dpi (Fig. 3g). The upregulated genes were mainly enriched in biological process terms, such as regulation of cell morphogenesis and axonogenesis, while the downregulated genes were primarily related to neuronal physiological functions, such as regulation of membrane potential, synapse organization or encoding ion channel (Supplementary Fig. 5c, d). The genes enriched in axonogenesis terms did not exhibit obvious changes in the SL (Fig. 3h). In contrast, there was an obvious increase in upregulated genes for neurons in the SL at 6 months (Fig. 3g), and GO analysis showed that these upregulated genes were enriched in biological processes, such as RNA splicing, regulation of protein catabolic process, and regulation of autophagy (Supplementary Fig. 5e, f). These findings suggest that regeneration genes were partially activated in neurons near the lesion during the acute phase, while neurons in distal tissues exhibited a weak response during the early phase but underwent autophagy during the chronic phase.

## Production of new subsets of OPCs and oligodendrocytes in both the proximal and distal spinal cord following SCI

We conducted further cluster analysis of oligodendrocyte lineage cells and identified 11 distinct subclusters, including three OPC clusters (*PCDH15*) and eight oligodendrocyte clusters (*ST18*) (Fig. 4a, b). Clusters 6, 7, and 8 were newly emerging clusters after injury (Fig. 4c). Based on gene expression, we named clusters 4, 6, and 9 the homeostatic OPCs (H-OPC), activated OPCs (A-OPC), and dividing OPCs (D-OPC), respectively. The proportions of D-OPC (9.1%) and A-OPC (49.4%) among OPCs in the SA were the highest at 7 dpi and subsequently decreased (Fig. 4d). D-OPC and A-OPC were also found in SL.

DEGs analysis revealed that A-OPCs downregulated homeostatic genes while upregulated genes involved in immune response and regulation of response to wounding (Supplementary Fig. 6a, b). Further analysis of DEGs between SA and SL showed that A-OPCs expressed higher levels of injury-induced genes (*GFAP*, *SERPINA3*) in SA but expressed more homeostatic genes (*GRM5*, *DSCAM*) in SL (Supplementary Fig. 6c, d). Additionally, H-OPCs also exhibited gene variation after SCI (Supplementary Fig. 6e). H-OPCs upregulated the expression of *GFAP* and *SERPINA3* in SA but at a lower level in SL (Supplementary Fig. 6f). We observed that the downregulated genes

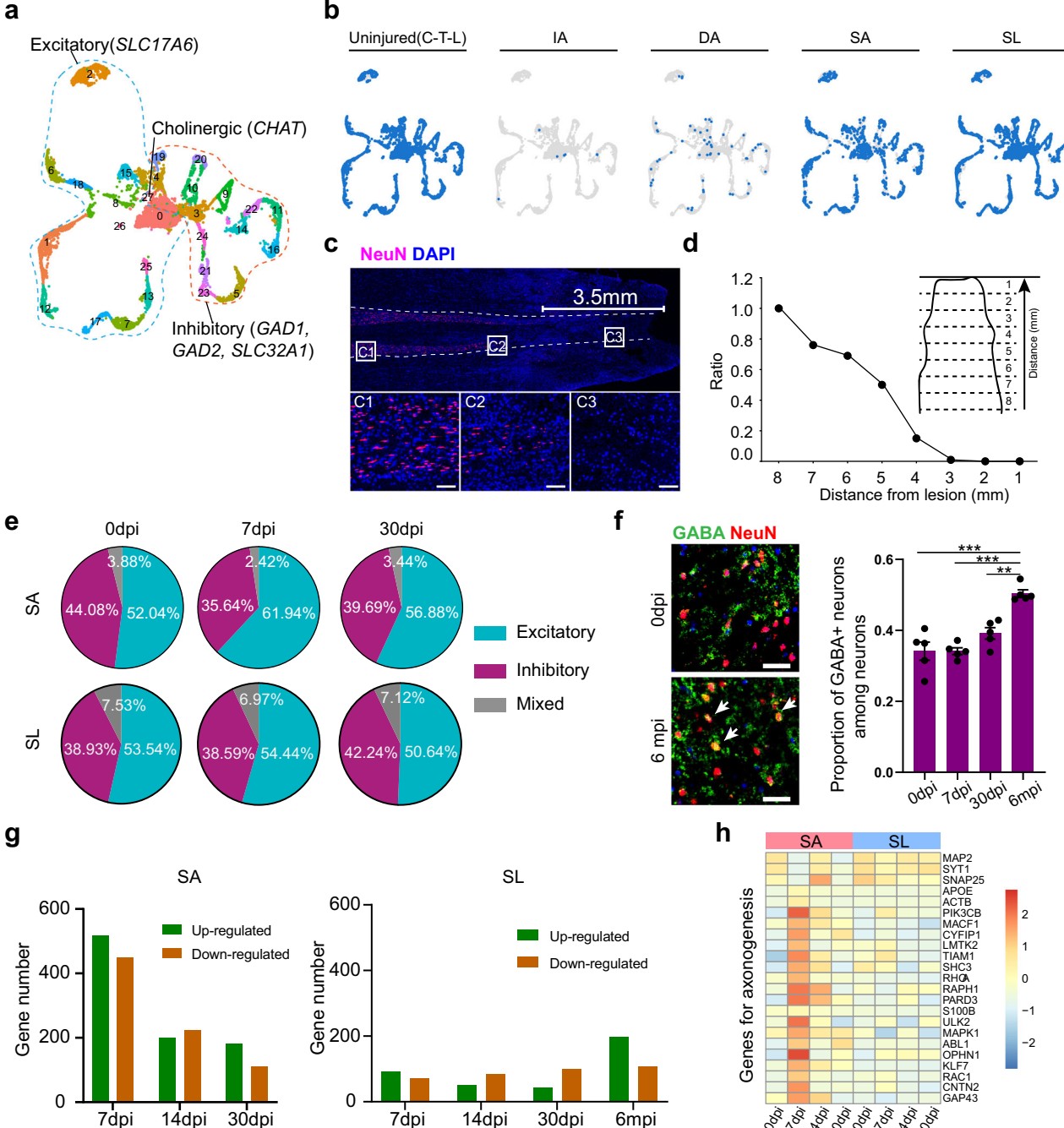

**Fig. 3 | Heterogeneous responses of neurons in the proximal and distal regions below the lesion after SCI. a** UMAP plot depicting twenty-eight subclusters of all neurons in the intact and injured spinal cord. The dashed lines indicate the neurotransmitter status. **b** Split UMAP plots depicting the harvested neuronal subsets in the uninjured spinal cord and different regions of the injured spinal cord. **c** Immunostaining of NeuN showing the distribution of survivable neurons in the injured rhesus monkey spinal cord at 7 dpi. The white scale labels the region with severe neuronal death. The dashed lines indicate the gray matter areas. The area in the white box is magnified. Scale bars in C1-C3, 200 μm. **d** The ratio of neuron number in different regions away from the injury site. **e** Pie plots showing the proportion of neurons with different neurotransmitters in the SA and SL area.

**f** Immunostaining of GABA and NeuN in the lumbar spinal cord (left) and bar plot showing the proportion of GABA⁺ neurons among neurons at each phase after SCI (right). Scale bars, 100 μm. Data are shown as mean ± SEM, $n = 5$ slices. $**p = 0.0011$, $***p < 0.0001$, one-way ANOVA coupled with Tukey's post hoc test. **g** Bar plot showing the number of upregulated genes and downregulated genes of neurons in SA and SL area after SCI. **h** Heatmap showing the relative expression of axonogenesis genes (from top GO term) in neurons of SA and SL area. SYT1, MAP2, and SNAP25 were used as marker genes to indicate neuron identity. The color bar is scaled with the average expression of the corresponding genes. Source data are provided as a Source Data file.

were enriched in the GO term of extracellular matrix (ECM) organization, and they were downregulated in SA but not in SL after SCI (Supplementary Fig. 6g). These findings suggest that although OPCs were activated from the proximal to distal regions after SCI, the response was weaker in SL.

## Generation of disease-associated oligodendrocytes following SCI

In the uninjured spinal cord, oligodendrocytes have six continuous subclusters (Fig. 4c and Supplementary Fig. 7a). Cluster 10 represents immature myelin-forming oligodendrocytes (MFOLs) derived from

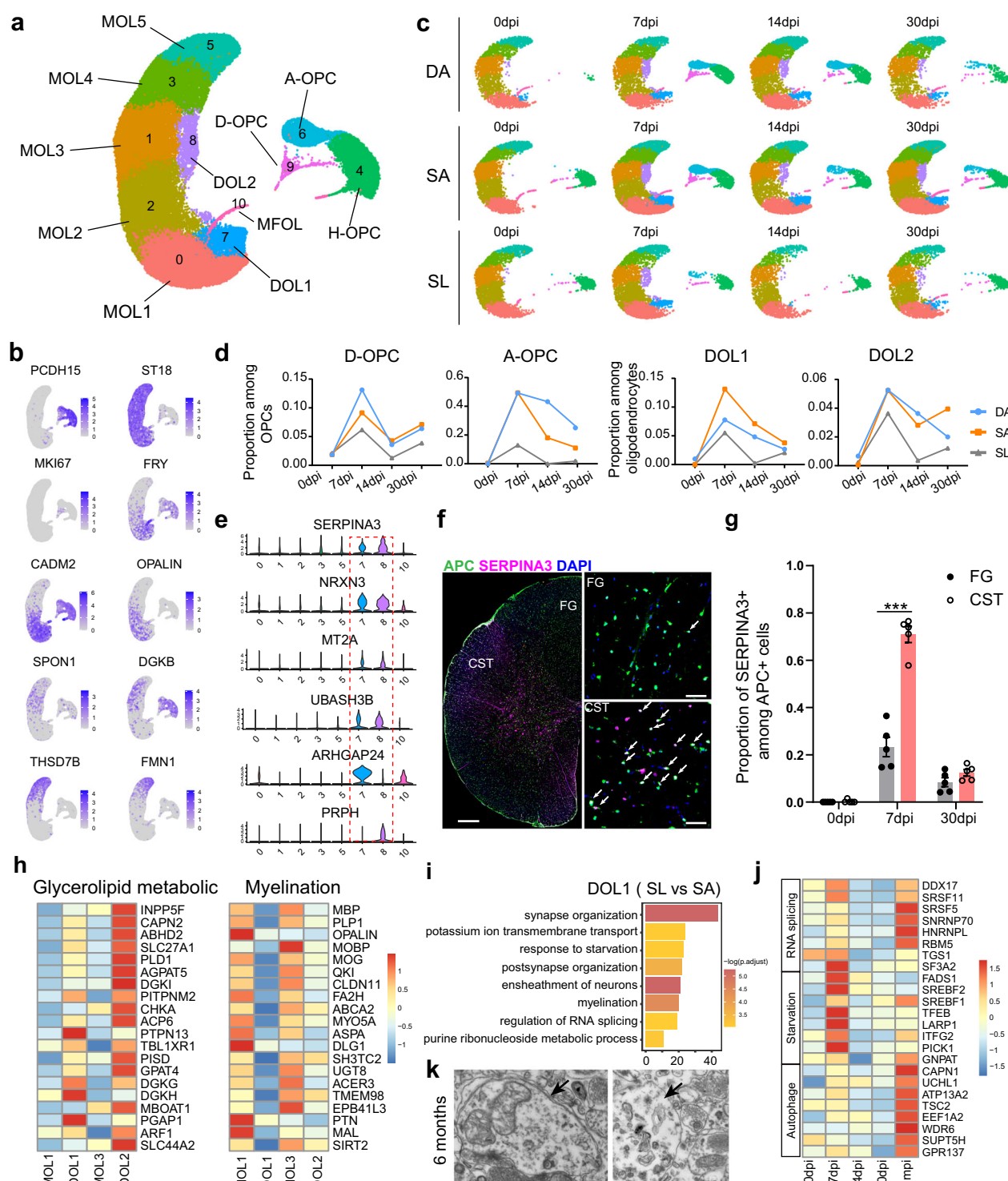

**Fig. 4 | Characteristics of injury-induced disease-associated oligodendrocytes. a** UMAP plot depicting the heterogeneous subsets of OPCs and oligodendrocytes in the intact and injured rhesus monkey spinal cord. A-OPC activated OPCs, H-OPC homeostatic OPC, D-OPC dividing OPC, MOL mature oligodendrocytes, MFOL myelin-forming oligodendrocytes, DOL disease-associated oligodendrocytes. **b** Gene expression visualized by UMAP plots. Each dot represents an individual cell colored according to the expression level. **c** Split UMAP plots showing the regional distribution of OPCs and oligodendrocyte subsets at different phases after SCI. **d** Dynamic changes in the proportion of subsets among OPCs or oligodendrocytes in different regions after SCI. **e** Violin plots showing the signature genes of DOL1 and DOL2 compared with other oligodendrocytes. **f** Immunostaining of SERPINA3 and APC in the lumbar spinal cord showing the different distribution of DOLs in CST and FG area at 7dpi. Scale bars, 500 μm on the left and 100 μm on the right.

**g** Bar plot showing the proportion of SERPINA3 positive oligodendrocytes at different time points after SCI. Data are shown as mean ± SEM, $n = 5$ slices. ***$p < 0.0001$, two-sided Student's $t$ test. **h** Heatmap showing the expression of genes related to glycerolipid metabolic process and myelination in oligodendrocyte subsets. The color bar is scaled with the average expression of the corresponding genes. **i** Enriched GO terms of the upregulated genes in DOL1 from the SL compared with that from the SA area. $P$ values (adjusted) were calculated using Benjamini–Hochberg false discovery rate (FDR). **j** Heatmap showing the expression of genes related to cellular stress in MOL1 from the SL. The color bar is scaled with the average expression of the corresponding genes. **k** Electron micrographs showing the autophagosome and autolysosome in the lumbar tissues at 6 months after SCI. Scale bars, 0.5 μm. Source data are provided as a Source Data file.

OPCs. We named another five clusters as mature oligodendrocytes 1–5 (MOL1–5). The top-ranked DEGs of each cluster exhibited a continuous transition gene expression pattern. Pseudo-time analysis reconstructing the lineage relationships showed that the oligodendrocyte lineages were distributed along pseudo-temporally ordered paths from OPCs to MOL5 (Supplementary Fig. 7a), indicating that oligodendrocytes in the spinal cord were in different stages of maturation.

Clusters 7 and 8 were newly emerging mature oligodendrocyte subtypes after SCI. They highly expressed *SERPINA3*, *NRXN3*, *MT2A*, and *UBASH3B* compared with other oligodendrocytes (Fig. 4e). *SERPINA3* has been reported as a key marker of disease-associated oligodendrocytes (DOLs) in neurodegenerative and autoimmune inflammatory conditions[40,41]. Therefore, we named these two clusters DOL1 and DOL2, respectively. We confirmed the distribution of these DOLs by immunostaining of SERPINA3 in SL. We found that SERPINA3⁺ oligodendrocytes were mainly distributed in the CST area and less in the FG area, accounting for about 70.9% in the CST area and 23.3% in the FG area among APC-positive oligodendrocytes at 7dpi (Fig. 4f, g). GO analysis showed that DOL1 and DOL2 upregulated genes shared many biological processes, such as synapse organization and tissue migration. However, DOL1 expressed more genes enriched in the regulation of angiogenesis, whereas DOL2 expressed more genes responding to interferon-gamma (Supplementary Fig. 7b). The correlation analysis demonstrated that DOL1 and DOL2 had similar gene expression patterns to MOL1 and MOL3, respectively (Supplementary Fig. 7c). Pseudo-time analysis of DOL1 and DOL2 with MOL1 and MOL3 showed that both DOL1 and DOL2 were located in the end branches of the trajectories (Supplementary Fig. 7d), suggesting that the DOL1 and DOL2 clusters represent responses to injury of different oligodendrocyte subtypes.

DEGs analysis of DOL1 and DOL2 compared with MOL1 and MOL3 respectively, showed that DOL1 and DOL2 shared many upregulated genes. However, DOL2 expressed more genes involved in lipid metabolism (Fig. 4h and Supplementary Fig. 7e, f). It was worth noting that the downregulated genes were enriched in biological process terms such as axon ensheathment and myelination in both DOL1 and DOL2. Correspondingly, the myelin-associated genes such as *PLP1*, *MBP*, and *MOG* were downregulated (Fig. 4h). These results suggested that DOLs had lower myelination genes expression compared with their correspondence MOLs, but further research is needed to determine their function.

### Cellular Stress in oligodendrocytes below the lesion following SCI

We investigated oligodendrocytes in SA and SL exhibited differences in gene expression. Similar to OPCs, the newly emerging DOLs also had region-related DEGs. Interestingly, we found that the upregulated genes of DOLs in SL were enriched in biological processes such as response to starvation and RNA splicing (Fig. 4i), which were similar to microglia and neurons in the SL. We examined the expression of genes involved in the regulation of RNA splicing, response to starvation, and regulation of autophagy in MOL1 and MOL3 in SL. The results revealed obvious upregulation of genes responding to starvation during the acute phase (7 days) and that genes related to the regulation of autophagy during the chronic phase, while genes related to RNA splicing were expressed at both the acute and chronic phases (Fig. 4j). This finding is consistent with the reports that the regulation of RNA splicing could allow rapid changes in gene expression in response to amino acid starvation in yeast[42–45]. Transmission electron microscopy (TEM) analysis showed the presence of autophagosome and autolysosome-like structures in the distal lumbar segments at 6 months after SCI (Fig. 4k). These results suggested that cells in the SL may experience cellular stress after SCI, including oligodendrocytes.

### Distinct gene signatures and functions of astrocytes in white matter and gray matter

Preliminary clustering analysis of cells in the uninjured spinal cord revealed two adjacent astrocyte clusters (7 and 14; Fig. 1e). Cluster 7 specifically expressed *VCAN* and *IL33*, while cluster 14 highly expressed *SLC1A2* and *ITPRID1* (Fig. 5a). Immunostaining of SLC1A2 revealed these two clusters represent white matter astrocytes (WM-AS) and gray matter astrocytes (GM-AS), respectively (Fig. 5b). IL33 is a well-recognized cytokine that is essential for immune regulation and inflammatory responses[46,47]. In adult mice, IL33 is reported to be mainly expressed in oligodendrocytes and astrocytes. However, IL33 was only expressed in WM-AS in the rhesus monkey spinal cord and 26-week-old human embryo spinal cord (Fig. 5c). These results demonstrate differences in gene expression between mice and primates. We performed cluster analysis of cells that express *IL1RL1*, which encodes a receptor of IL33, and found the expression of *IL1RL1* was higher in T cells (Supplementary Fig. 8a, b). We speculated that the release of IL33 from WM-AS after SCI might play an important role in lymphocyte recruitment.

### Different responses of astrocytes in white matter and gray matter following SCI

To analyze the heterogeneity of astrocytes after SCI, we performed further cluster analysis of all astrocytes and identified eight distinct subtypes (Fig. 5d). Clusters 1, 6, and 7 were newly emerging subtypes after SCI (Fig. 5e). Clusters 0, 1, 2, 4, and 5 expressed markers of white matter astrocytes (*IL33*), while clusters 3 and 6 expressed markers of gray matter astrocytes (*ITPRID1*). We therefore labeled clusters 1 and 6 as activated white matter astrocytes (AW-AS) and activated gray matter astrocytes (AG-AS), respectively. Cluster 7 was dividing astrocytes(D-AS) that expressed cell-cycle genes (*MKI67*, *TOP2A*) (Fig. 5f).

Astrocytes can be polarized in a destructive orientation (A1) or protective orientation (A2) in mouse CNS disease models[48–50]. We examined the expression of the A1 (*C1r*, *C1s*, and *C3*) and A2 (*LIF*, SPP1, and *THBS1*) associated genes in astrocytes from DA and SA. Interestingly, we found that AW-AS and AG-AS expressed high levels of A2 genes and low levels of A1 genes, whereas other WM-AS expressed relatively high levels of A1 genes (Supplementary Fig. 8c).

The percentage of AW-AS among astrocytes remained at a high level in SA from the acute to chronic phase after SCI. However, the percentage of AW-AS in SL reached a peak at 7 dpi and decreased sharply thereafter. Similarly, the percentage of AG-AS reached a peak at 7 dpi and then almost disappeared in all regions at 30 dpi (Fig. 5g). We found AW-AS expressed more genes enriched in the regulation of vasculature development, while AG-AS expressed a high level of genes involved in the regulation of trans-synaptic signaling (Fig. 5h). The high percentage and the regulation of the vasculature development function of WM-AS may play more important roles in scarring and angiogenesis after SCI.

*NESTIN* has been used as a marker of neural stem cells during development and has been reported to be reactivated in multiple cell types in the injured spinal cord[51,52]. We re-clustered the *NESTIN*-expressing cells and showed that astrocytes, OPCs, oligodendrocytes, ependymal cells, endothelial cells, fibroblasts, and pericytes expressed *NESTIN* after SCI (Supplementary Fig. 8d). Next, we integrated the *NESTIN*⁺ cells with human embryonic spinal cord cells (7 weeks gestation) that include neural precursor cells (NPCs)[21]. Interestingly, we found partial astrocytes of rhesus monkey were clustered with NPCs of human embryonic spinal cord (Supplementary Fig. 8e), indicating that these astrocytes had similar gene expression with human spinal cord NPCs. The astrocytes had two subclusters (clusters 2 and 16), and cluster 2 was more similar to human NPCs. DEGs analysis showed that cluster 2 highly expressed *PAX3* and *ASCL1*(Supplementary Fig. 8f), which are neural stem cell-associated genes[53,54]. Further analysis demonstrated that these cells expressed marker genes of white matter

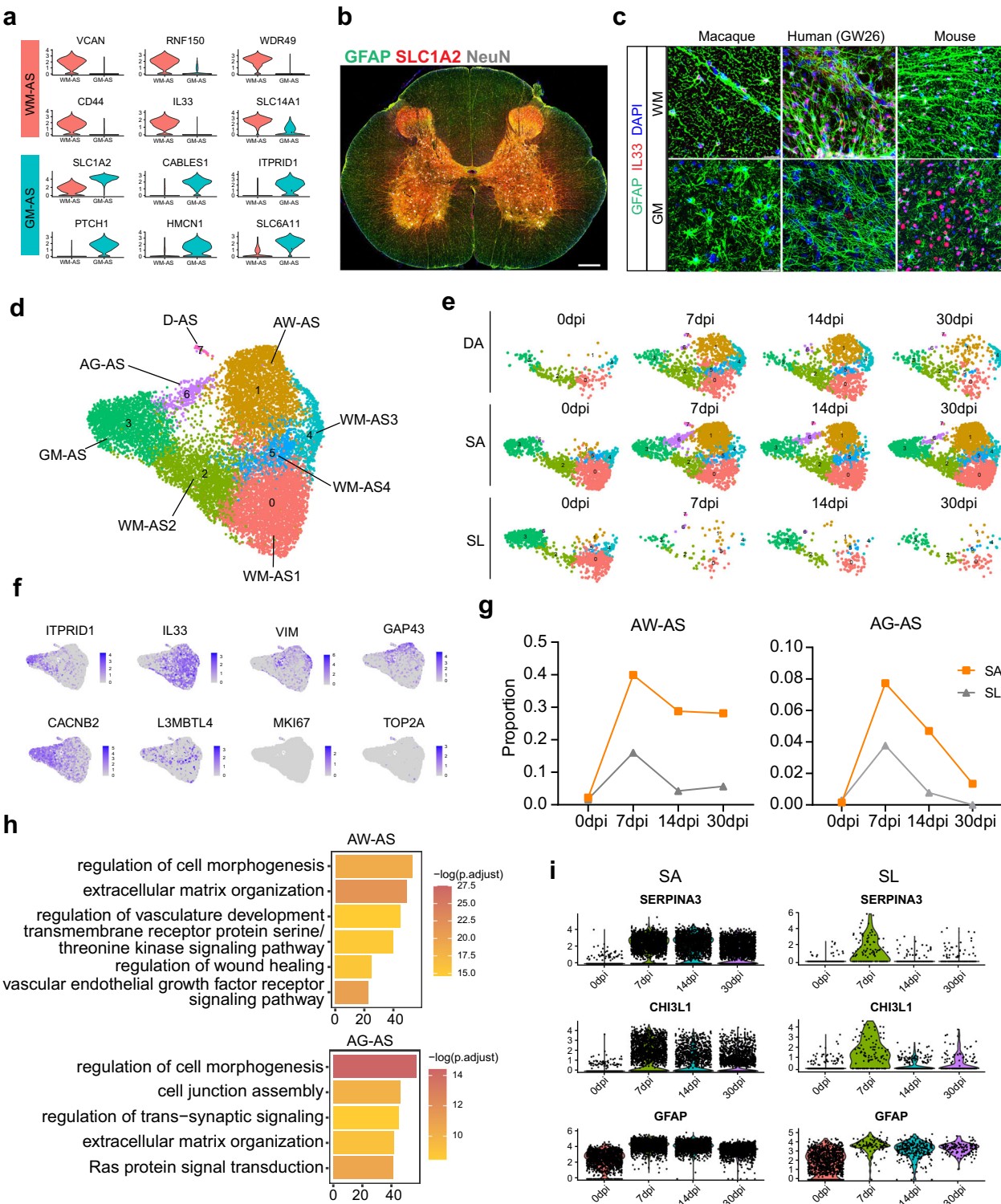

**Fig. 5 | White matter astrocytes and gray matter astrocytes respond to injury differently and are slightly activated in the distal regions after SCI. a** Violin plots showing the differentially expressed genes between white matter astrocytes (GM-AS) and gray matter astrocytes (WM-AS). **b** Immunostaining of SLC1A2, GFAP, and NeuN in the rhesus monkey spinal cord cross-section. Scale bars, 500 μm. **c** Immunostaining of GFAP and IL33 in the spinal cord cross-sections of rhesus monkey, human embryo (26 weeks), and mouse. Scale bars, 25 μm. **d** UMAP plot depicting the heterogeneity of astrocytes in the intact and injured spinal cord of rhesus monkey. WM-AS white matter astrocytes, AW-AS activated white matter astrocytes, GM-AS gray matter astrocytes, AG-AS activated gray matter astrocytes,

D-AS dividing astrocytes. **e** Split UMAP plots showing the regional distribution of astrocyte subsets at distinct time points after SCI. **f** Gene expression visualized by UMAP. Each dot represents an individual cell colored according to the expression level. **g** Dynamic changes in the proportion of AW-AS and AG-AS in different regions after injury. **h** GO enrichment analysis of the upregulated genes of AW-AS and AG-AS compared with WM-AS and GM-AS, respectively. *P* values (adjusted) were calculated using Benjamini–Hochberg false discovery rate (FDR). **i** Violin plots showing the expression of injury-associated genes in astrocytes from SA and SL area. Source data are provided as a Source Data file.

astrocytes but not gray matter astrocytes (Supplementary Fig. 8g). These results further reveal the possible plasticity of white matter astrocytes in the injured spinal cord.

Previous reports in mice identified *Serpina3n* as a marker of reactive astrocytes[55]. Our data showed that all astrocytes in the SA showed upregulation of *SERPINA3*, *CHI3L1*, and *GFAP*. However, most of the injury-induced genes except *GFAP* were upregulated in the SL mainly at 7 dpi and recovered to lower levels thereafter (Fig. 5i). Combined with the low proportion of AW-AS and AG-AS in SL, these results suggest the astrocytes were minimally activated in the distal lumbar tissue after SCI.

### Conserved cellular responses below the lesion of rhesus monkeys and mice after SCI

We integrated our data with the published data on the lumbar spinal cord of mice to better understand the similarities and differences between mice and rhesus monkeys[8]. The integration of whole cells showed that rhesus monkeys and mice spinal cord had similar cell composition and dynamics after SCI (Supplementary Fig. 9a, b). We conducted further analyses of neurons, microglia, and oligodendrocytes. The integration results here demonstrated the conservation of excitatory-inhibitory and dorsal-ventral clustering patterns between macaque and mouse lumbar spinal cord neurons (Supplementary Fig. 9c, d). Similarly, we observed an increase in the proportion of inhibitory neurons from 45.21% to 50.88% in the lumbar region of mice 3 weeks after SCI (Supplementary Fig. 9e). Rhesus monkeys and mice lumbar tissues shared similar subtypes of microglia, including homeostatic microglia, DAMLM, inflammatory microglia, and proliferative microglia (Supplementary Fig. 9f, g). However, we found differences in their subsets distribution. A higher proportion of inflammatory microglia, DAMLM, as well as dividing microglia were seen in rhesus macaque after SCI, whereas mice had more homeostatic microglia (Supplementary Fig. 9h, i). Rhesus monkeys and mice both developed disease-associated oligodendrocytes in lumbar tissue after SCI, but we found more A-OPC in rhesus macaques and more DOLs in mice (Supplementary Fig. 9j–l). These results suggested that cells below the lesion in rhesus macaques and mice exhibit overall conserved cellular responses after SCI, but the differences may be related to species and the different SCI models (crush or transection).

### Intense cell interactions in proximal regions after SCI mediated by multiple scar-forming cells

We analyzed the cell–cell interactions of different regions after SCI. First, we conducted a global survey of the cellular interactions between the cells in DA and SL. We found that the cellular interactions were enhanced in both DA and SL at 7 dpi. For example, the number of ligand-receptor pairs between fibroblasts and endothelial cells increased from 210 to 325, and ligand-receptor pairs between fibroblasts and OPCs increased from 106 to 238 in DA. Ligand-receptor pairs between neurons and endothelial cells increased from 174 to 217 in SL (Supplementary Fig. 10a–f). Cellular interactions almost returned to pre-injury level by 30 dpi in SL but not in DA, as demonstrated by the interactions between OPCs and other cells (Supplementary Fig. 10c).

Considering fibrotic and glial scar tissues were mainly composed of fibroblasts, endothelial cells, astrocytes, OPCs, and immune cells, we evaluated the interaction between these scar-forming cells at 7dpi in DA. We found strong interactions between endothelial cells and fibroblasts. In addition to canonical vascular signaling pathways VEGFA-FLT1, fibroblast ligands like FN1 and collagen were involved in the interaction with endothelial cells (Supplementary Fig. 10g). Microglia via SPP1, CD74, and CD44 interacted with multiple cell types, including endothelial cells, astrocytes, and OPCs (Supplementary Fig. 10g–i). Astrocytes interacted with multiple cells via FGFR1-NCAM1, FGF2-CD44, and OPCs interacted with various cells via PTN-PTPRZ1 and NRP2-SEMA3C. Notably, a strong SPP1-CD44 signaling between

astrocytes and microglia/macrophage as well as oligodendrocytes was observed, which was similar in OPCs and microglia/macrophage. Expression analysis showed that SPP1 was mainly expressed in DAMLM microglia but not in macrophages (Supplementary Fig. 10j). We noted that T cells interacted with endothelial cells, microglia, and OPCs via TGFB1-TGFBR2. TGF-β signaling is critical for the injury/repair response and astrogliosis[56]. Further analysis showed that *TGFBR2* was primarily expressed in activated astrocytes (AW-AS and AG-AS) and dividing astrocytes (Supplementary Fig. 10k). These results reveal that intense interactions between different cell subtypes play important roles in pathological changes in proximal regions after spinal cord injury.

### Scaffold implantation improves the microenvironment of the injury area and promotes axon regeneration

Previous studies showed that implantation of a functional collagen scaffold could improve the pathological and motor function restoration in rats, dogs, and primates with severe SCI[14,16,57–60]. Here, we analyzed the potential effects of linear-ordered collagen scaffold (LOCS) implantation loaded with EGFR antibody cetuximab (CTX) for SCI in monkeys. Similar to previous results, we observed obvious axon regeneration in the injury area at 6 months after scaffold implantation (Fig. 6a, b). The scRNA-seq results at 6 months showed that the injury area comprised of stromal cells (fibroblasts, endothelial cells, and pericytes), immune cells (microglia, macrophages, T cells, and neutrophils), and neuroglia cells (oligodendrocytes, OPCs, and Schwann cells) (Fig. 6c). Interestingly, the proportion of glial cells, such as OPCs, oligodendrocytes, and Schwann cells increased after scaffold implantation (Fig. 6c–g). Glial cells can secrete various of neurotrophic factors to facilitate axonal regeneration or remyelination after SCI[57,61,62]. We found cells after LOCS + CTX implantation expressed higher levels of neurotrophic factors. Specifically, most Schwann cells expressed a high level of *FGF1* (Fig. 6h). GO analysis results also showed that Schwann cells expressed more genes involved in axonogenesis, myelination, and axon guidance (Fig. 6i).

Fibroblasts are the major cells that secrete many extracellular matrix (ECM) components to seal the injury sites[7,63,64]. DEGs analysis showed differentially expressed genes of fibroblasts between SCI and LOCS + CTX groups. Fibroblasts after scaffold implantation expressed more genes for positive regulation of neurogenesis and axon guidance (Supplementary Fig. 11a, b). There were mainly two fibroblast subsets, and the proportion changed after scaffold implantation (Supplementary Fig. 11c–e). Perivascular fibroblasts exclusively expressed ABC efflux pumps, whereas meningeal fibroblasts specifically expressed SLC influx solute transporters[65]. We checked the gene expression and identified clusters 0 and 1 to represent meningeal and perivascular fibroblasts, respectively (Supplementary Fig. 11f). Perivascular fibroblasts expressed higher levels of growth factors while lower levels of proteoglycans such as CSPGs, which have been reported attenuate axonal growth after SCI (Supplementary Fig.11g, h)[66,67]. We verified this by immunostaining of VCAN in the injury sites (Supplementary Fig. 11i, j). These results suggest scaffold implantation could regulate the fibroblast properties and matrix composition in the lesion sites.

### Scaffold implantation improves microenvironment in the distal lumbar spinal cord after SCI

We found functional scaffold implantation could substantially prevent the increase of inhibitory neurons in the spared lumbar (Fig. 7a). We confirmed this by immunostaining of GABA in the lumbar spinal cord sections, which showed a decreased proportion of GABA-positive neurons in the dorsal horn and few GABA-positive signals surrounding neurons of the ventral horn (Fig. 7b–d). We further found that neurons upregulated genes related to the regulation of trans-synaptic signaling, synapse organization, and regulation of membrane potential after LOCS + CTX implantation (Fig. 7e). Go analysis also showed neurons in

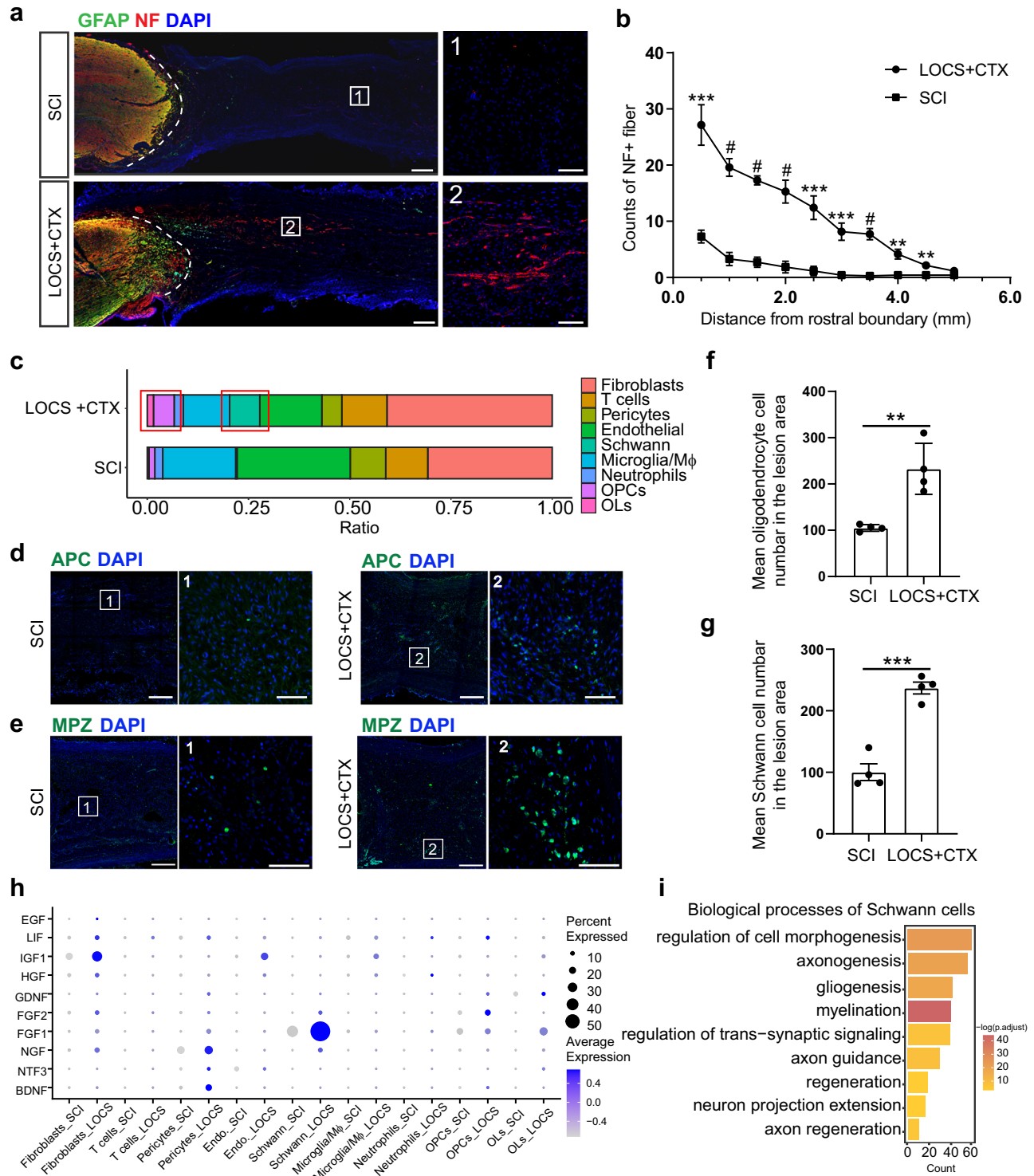

LOCS + CTX group expressed higher levels of genes for regulation of trans-synaptic signaling, synapse organization, as well as molecular function related to ion channel activity and glutamate receptor activity (Fig. 7f, g). These results suggest treatment with a functional scaffold could partially rescue the dysfunctions of neurons in the distal lumbar after SCI.

We also examined the expression of neurotrophic factors and growth factors in lumbar cells (Fig. 8a). Compared to SCI group without scaffold implantation, cells in LOCS + CTX group highly expressed *BDNF* and *NT3* (Neurons), *FGF1*, *FGF2*, and *EGF* (Astrocytes), *HGF* (Microglia), and *IGF1* (Endothelial cells). Further analysis showed that

oligodendrocytes expressed higher levels of myelination genes after scaffold implantation (Fig. 8b, c). Inflammatory microglia decreased in the scaffold implantation group (Fig. 8d, e). Thicker and denser myelin sheath structures and fewer TNF-α positive signals were seen in the lumbar spinal cord after scaffold implantation (Fig. 8f–h). Additionally, we found the DAMLM microglia expressed higher levels of genes related to phagocytosis and lipid metabolism after LOCS + CTX implantation (Fig. 8i). Immunostaining with GPNMB and lipid droplet staining with BODIPY showed a higher expression level of GPNMB and fewer lipid droplets in the spinal cord with scaffold implantation (Fig. 8j, k), suggesting the better lipid scavenging ability. These results

**Fig. 6 | Functional scaffold implantation promotes axon regeneration and improves the microenvironment of the injury area. a** Immunostaining of NF and GFAP showing the axonal regeneration in the injury area six months after SCI and scaffold implantation. The areas in the white boxes are magnified. SCI and LOCS + CTX represent injured spinal cord without or with scaffold implantation respectively. Scale bar, 500 μm on the left and 100 μm on the right. **b** Quantification of the number of NF-positive neural fibers along the distance from the rostral boundary of the injury area in two groups. Data are shown as mean ± SEM, *n* = 7 per group. **p = 0.0015 (4.0 mm); 0.0051 (4.5 mm), ***p = 0.0002 (0.5 mm); 0.0003 (2.5 mm); 0.0004 (3.0 mm), #p < 0.0001 (1.0 mm, 1.5 mm, 2.0 mm, 3.5 mm), two-sided Student's *t* test at each distance. **c** Bar plot showing the cellular proportion in the injury area six months after SCI and scaffold implantation. **d** Immunostaining for APC showing the distribution of oligodendrocytes in the injury area at six months. The areas in the white boxes are magnified. Scale bar, 500 μm in each left

panel, 50 μm in panels 1 and 2. **e** Immunostaining for MPZ showing the distribution of Schwann cells in the injury area at six months. The areas in the white boxes are magnified. Scale bar, 500 μm in each left panel, 100 μm in panels 1 and 2. **f, g** Quantitative analysis showing the mean number of oligodendrocytes (APC$^+$) and Schwann cells (MPZ$^+$) in the injury area. Data are shown as mean ± SEM, *n* = 4 slices per group. **P = 0.0037, ***P = 0.0002, two-sided Student's *t* test. **h** Dot plot showing the expression of growth factors in the injury area at six months. The size of the dot indicates the percentage of cells in which that gene is detected and the color bar is scaled with the average expression of the corresponding genes. **i** GO enrichment analysis of the differentially expressed genes of Schwann cells compared with other cells. *P* values (adjusted) were calculated using Benjamini–Hochberg false discovery rate (FDR). Source data are provided as a Source Data file.

suggested the implantation of LOCS + CTX may also improve the efficiency of microenvironment remediation in the distal lumbar by promoting microglial-mediated debris phagocytosis and lipid recycling.

## Discussion

We constructed a cell atlas of different regions in the injured spinal cord of rhesus monkey from the acute to chronic phases (Fig. 9). The atlas depicted the unique molecular heterogeneity and the spatio-temporal dynamics of multiple cell types in the injured rhesus monkey spinal cord. Specifically, we revealed cells in the distal lumbar tissues below the lesion were also affected and generated a degenerative microenvironment. The fibrosis, gliogenesis, and inflammatory responses to SCI decreased with the distance from the injury site, but genes related to cellular stress, phagocytosis, and autophagy were upregulated below the lesion. Furthermore, we discovered that functional scaffold implantation could improve the microenvironment of both the lesion sites and the spared lumbar, and rescue the dysfunction of neurons below the lesion. These findings may provide insights into potential targets below the lesion for SCI repair.

Microglia, as the resident macrophages of the CNS, have important physiological functions such as immune monitoring, apoptotic cell clearance, and synaptic pruning[68]. Our data showed that microglia responded strongly to injury and were induced into a state resembling DAM, which was responsible for clearing cell fragments in the descending tract region below the lesion. Previous reports have suggested microglia with neuroprotective properties can be beneficial for functional recovery in the early stage of SCI[69]. Here, we found that DAMLM had low expression of pro-inflammatory M1 genes. The protective function of DAM in neurodegenerative diseases suggests the beneficial roles of DAM[34,35], but whether they are beneficial for SCI repair needs further research.

After SCI, external immune cells infiltrate the injured area and mediate a strong inflammatory response[68,70]. In our complete SCI model, we found the areas that were implicated far exceeded the actual area of damage. Cells within ~3 mm of both ends of the lesion underwent massive apoptosis. Interestingly, we found neurons proximal to the lesion increased expression of genes related to axonogenesis at 7 dpi and decreased thereafter, which was similar to previous reports in mouse[71]. It highlights the importance therapeutic window for neural regeneration in the acute phase after SCI.

Previous studies reported different responses of oligodendrocytes to SCI in mice[28]. Similarly, we showed that oligodendrocytes of rhesus monkeys in different stages of maturity respond variably to SCI. There were two emerging disease-associate oligodendrocyte subsets, DOL1 and DOL2, which had similar gene expression patterns to MOL1 and MOL3, respectively. The SERPINA3+ DOLs were primarily found in the CST area, similar to DAMLM. The CST (corticospinal tract) is a major spinal tract responsible for transmitting signals from the cerebral cortex to the spinal cord. The cell bodies of these neurons are located in the cerebral cortex. In our complete spinal cord injury (SCI)

model, we believe that the axons, including those in the CST, have been severed from their cell bodies, resulting in stereotypical Wallerian degeneration of the distal portion of the axons below the lesion, accompanied by degeneration of their associated myelin sheaths[72]. This degeneration of axons and myelin sheaths triggers the activation of microglia and other cellular responses.

Emerging evidence suggests that oligodendrocytes are also active participants in the neuroimmune network. Interferon factor has been reported to be the main transcription factor orchestrating oligodendrocyte demyelination in multiple sclerosis (MS) and experimental autoimmune encephalomyelitis (EAE) mice[73–76]. Here, we found that the emerging subsets showed downregulated expression of myelination-associated genes and upregulated genes involved in regulation of immune response. DOL2 cells showed upregulation of genes involved in the response to interferon-gamma. The positive or negative functions of DOL1 and DOL2 cells for SCI need to be investigated in future studies.

Our data revealed conservation in the cellular responses between monkeys and mice spinal cords. After SCI, the cells in the lumbar region of monkeys and mice exhibited similar injury responses, with an increased proportion of inhibitory neurons and the similar subtypes of microglia and oligodendrocytes. However, there were also differences in gene expression between monkeys and mice. For example, *IL33* was widely expressed in oligodendrocytes and astrocytes in mice[46,47], while it was exclusively expressed in white matter astrocytes in primate spinal cord. The proportions of different cell subtypes below the lesion were also distinct between the two species. The inflammatory response was stronger in monkeys, which may explain the different pathological reactions between mice and monkeys. However, It should be noted that the mouse injury model used by Matson et al. was a contusion[8], which is different from our complete transection injury model.

Considerable attention has been paid to the injury sites after SCI[6,7,26,72,77,78]. We found that cells, including neurons, oligodendrocytes, and microglia in the spared lumbar segments showed upregulation of genes involved in regulation of autophagy, RNA splicing, and response to starvation. In our complete spinal cord injury model, the spinal longitudinal arteries were completely broken, leading to the breakdown of the longitudinal blood supply in the spared lumbar tissue. It might partially explain the insufficient supply of nutrition to cells in the spared lumbar. These data suggest that the abnormal physiological activity of neurons and oligodendrocytes in SL might induce microglia into a phagocytic state responsible for the recycling of myelin debris after SCI.

Our previous results had shown the implantation of LOCS with cetuximab could effectively promote motor function restoration in rats and dogs after SCI[14]. In this study, we also observed obvious axon regeneration in the injury sites after implantation of LOCS loading cetuximab in monkeys. We found the functional scaffold could improve the microenvironment of injury sites by recruiting glial cells

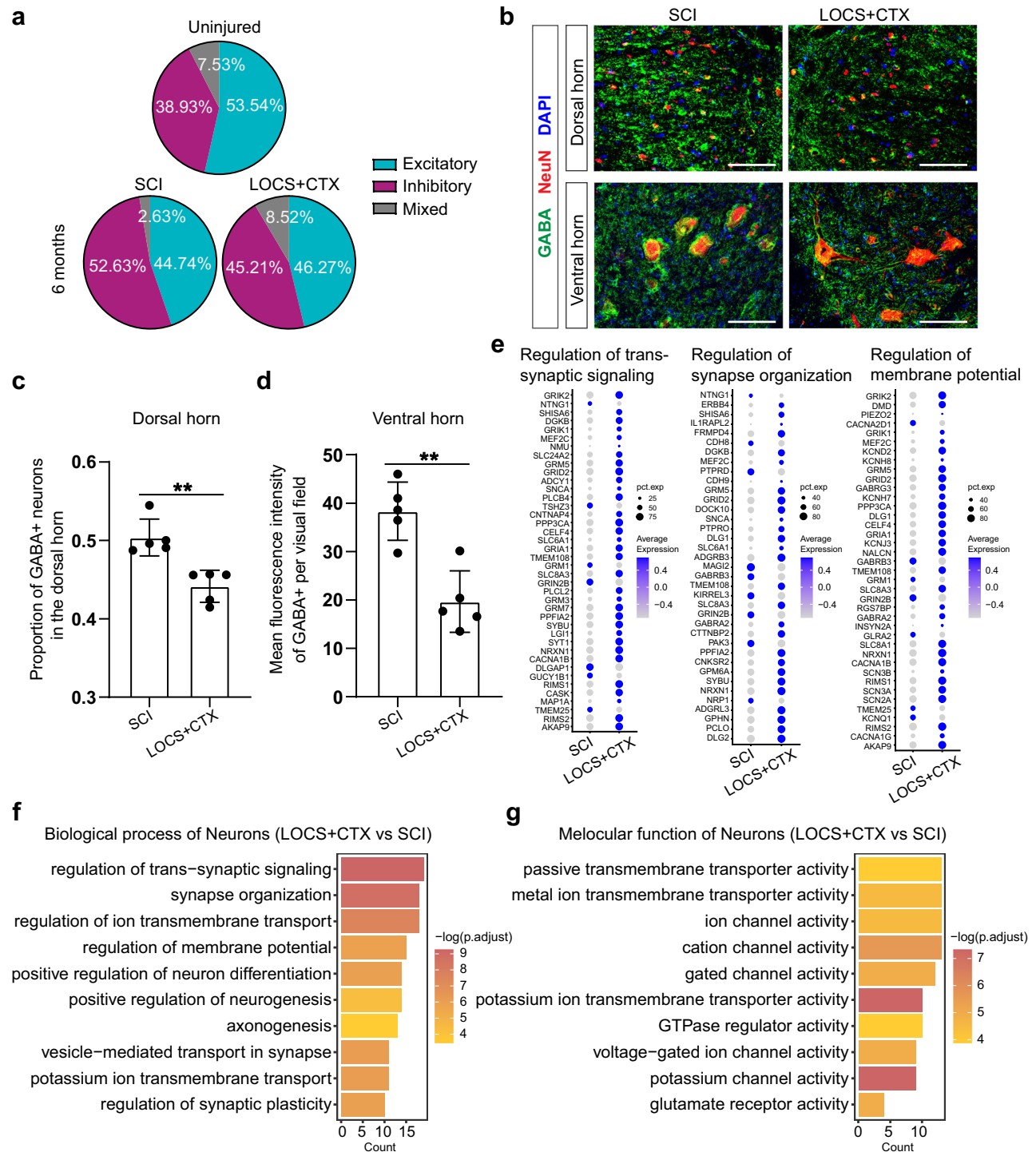

**Fig. 7 | Functional scaffold implantation decreases the secretion of inhibitory transmitters and maintains the synaptic functions of neurons in the lumbar spinal cords. a** Pie plots showing the proportion of neurons with different neurotransmitters at six months. **b** Immunostaining images showing the distribution of inhibitory neurotransmitters (GABA) in the lumbar six months after SCI and scaffold implantation. Scale bar, 100 μm. **c**, **d** Quantitative analysis showing the proportion of GABA-positive neurons in the dorsal horn and mean fluorescence intensity of GABA-positive signal in the ventral horn of the lumbar six months after SCI and scaffold implantation. Data shown as mean ± SEM, *n* = 5 slices per group.

**P = 0.0021 (**c**) ; 0.0014 (**d**), two-sided Student's *t* test. **e** Dot plots showing the expression of genes about the regulation of trans-synaptic signaling, synapse organization, and membrane potential in neurons of lumbar six months after SCI and scaffold implantation. The size of the dot indicates the percentage of cells in which that gene is detected and the color bar is scaled with the average expression of the corresponding genes. **f**, **g** Enriched biological process and molecular function of the upregulated genes of neurons in the lumbar six months after scaffold implantation. *P* values (adjusted) were calculated using Benjamini−Hochberg false discovery rate (FDR). Source data are provided as a Source Data file.

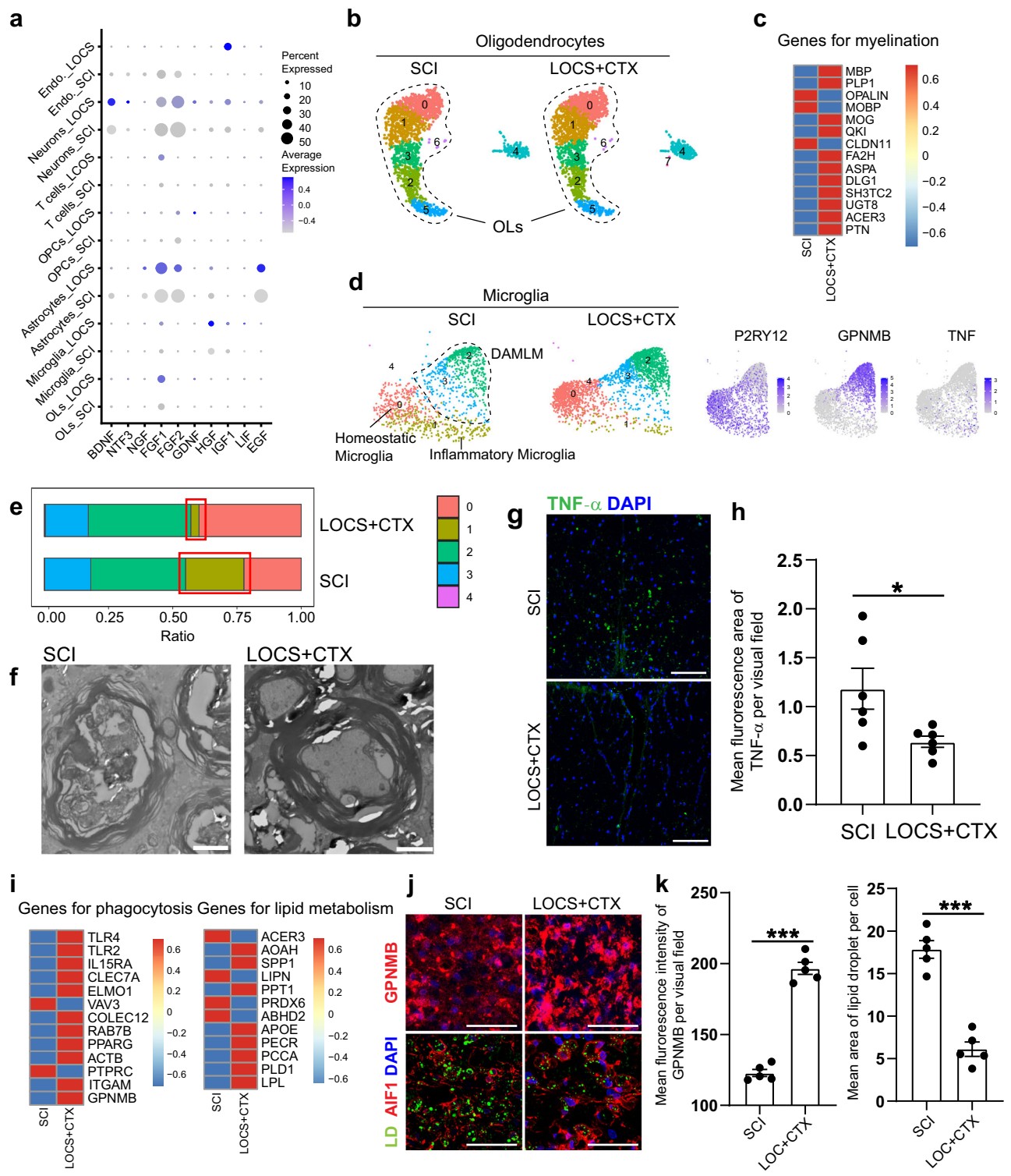

and perivascular fibroblasts and secreting more neurotrophic factors beneficial for neural regeneration. Additionally, we found that functional scaffold could mitigate the increase of inhibitory neurons induced by SCI and maintain the neuronal physiological functions in the lumbar spinal cord. Previous studies have suggested a substantially reduced excitability of the spinal network in the spared lumbar after SCI[10,79,80]. Considering the excitation-inhibition balance of circuits below the lesion is critical for locomotor recovery[11], the reduced inhibitory neurons after scaffold implantation may be critical to the recovery of motor function. However, due to the limitation of the sample size of monkeys in our study, these results still need further in-

depth research in the future. Nevertheless, our findings provide comprehensive insights for the development of combined repair strategies to promote regeneration of proximal tissue and to protect distal spinal cord tissue after SCI.

## Methods
### Animals
A total of 12 adult female rhesus monkeys (*Macaca mulatta*) aged 4–7 years were used in this study. Two monkeys were used as control without any treatment, while the remaining ten animals were used to establish the spinal cord injury (SCI) model. Two animals for 7 days,

**Fig. 8 | Functional scaffold implantation remodels the distal lumbar micro-environment. a** Dot plot showing the gene expression of growth factors in cells of the distal lumbar. The size of the dots represents the percent of cells expressing each gene, while the color depicts the scaled average expression. **b** UMAP plots depicting the oligodendrocyte subsets in the distal lumbar after SCI without or with scaffold implantation at six months. **c** Heatmap showing the expression of myelination genes in oligodendrocytes after SCI without or with scaffold implantation. The color bar is scaled with the average gene expression. **d** UMAP plots showing the microglia subsets in the lumbar after SCI without or with scaffold implantation and gene expression for homeostatic, DAMLM, and inflammatory microglia. **e** The proportion of microglia subsets after SCI with and without scaffold implantation at six months. The red frame indicating inflammatory microglia. **f** Electron micrographs showing the myelin sheath in the lumbar tissue six months after SCI without and with scaffold implantation. Scale bar, 2 μm. **g** Immunostaining shows the decreased expression of TNF-α in the distal lumbar tissues after scaffold implantation. Scale bar, 100 μm. **h** Quantitative analysis showing the mean fluorescence area of TNF-α positive signal in the same visual fields of distal lumbar tissues. Data are shown as mean ± SEM, $n = 6$ slices per group. *$P = 0.0318$, two-sided Student's $t$ test. **i** Heatmap showing the expression of genes about phagocytosis and lipid metabolism in DAMLM six months after SCI and scaffold implantation. The color bar is scaled with the average expression of the corresponding genes. **j** Immunostaining of GPNMB and lipid droplets (LD) labeled with BODIPY in the lateral CST area of the distal lumbar tissues six months after SCI and scaffold implantation. Scale bar, 100 μm. **k** Quantitative analysis showing the mean fluorescence intensity of GPNMB in the same visual fields and the mean area of lipid droplets per cell in the distal lumbar. Data are shown as mean ± SEM, $n = 5$ slices per group. ***$P < 0.0001$, two-sided Student's $t$ test. Source data are provided as a Source Data file.

14 days, and 30 days each, 4 animals for 6 months and 2 of them were used for scaffold implantation. Among them, one animal was used for scRNA-seq/snRNA-seq and the other one was used for pathological analyses for each group. All animals were supplied and housed at Beijing Institute of Xieerxin Biology Resource with accreditation of Laboratory Animal Care accredited facility and surgeries were performed there. All experimental and surgical procedures were in accordance with the Guide for the Care and Use of Laboratory Animals from the National Institutes of Health and approved by the Animal Care and Use Committee of Beijing Institute of Xieerxin Biology Resource (permit no. 20191017).

### Surgical procedures
The monkeys were anesthetized with 1 mg/kg ketamine intramuscularly and maintained under anesthesia with 1.5–2.5% isoflurane. While in the prone position, an intravenous drip infusion of physiological saline was administered. Blood oxygen saturation, heart rate, and respiratory rate were monitored throughout the surgery. Under the aseptic conditions, a longitudinal incision ~5 cm in length, centered at T9 was made. The erector spine muscle was dissected away from the spinous process and retracted laterally. The entire T9 dorsal lamina was removed using bone rongeurs. Then the dura was slit longitudinally along the midline with a surgical blade to expose about 1 cm of the spinal cord. Finally, an 8-mm-long block of the spinal cord in the center was cut and removed using iridectomy scissors and aspiration, with visual verification to ensure complete transection ventrally and laterally. After achieving complete hemostasis with geofoams, the dura, paraspinal muscles, and skin were closed in layers with sutures.

### Functional scaffold implantation
The linear-ordered collagen scaffold (LOCS) was prepared from bovine aponeurosis as described before[14]. The functional collagen scaffold, an 8 mm-long bundle of LOCS fibers was incubated by 100 μL cetuximab (CTX) (Merck Serono, S20130004) at room temperature for 30 min. After incubation, the functional collagen scaffolds were implanted into the lesion site.

### Postoperative care
The monkeys were given intravenous antibiotics and glucose daily for at least 3 days. The wound was cleaned with povidone-iodine every day. To prevent pressure ulcers, the monkey's skin was carefully examined twice a day. Manual bladder expression and gentle abdominal massage were performed to avoid urinary retention.

### Tissue harvesting
Considering the massive cell death around the injury sites, we performed single-cell sequencing for tissues in the injured area (IA) and degenerative area (DA) regions, and performed single-nucleus sequencing for spared activated area (SA) and spared lumbar area (SL) to acquire high-quality sequencing data. Animals were deeply anesthetized and transcardially perfused with prechilled saline (0.9%), and the spinal cord was dissected out of the spinal column. For histological analysis, a prechilled solution of 4% PFA was perfused, followed by saline. For single-cell/nucleus sequencing, the spinal cord meningeal was removed. The fibrotic scar (IA) has distinct characteristics with the spinal cord parenchymal tissue and was prepared for single-cell suspension. Five-millimeter-long spinal cord tissues next to the IA (DA) were prepared for single-cell (0, 7, 14, 30 days) or nucleus (6 months) suspension. Spinal cord tissues in SA, another 5 mm in length spinal cord tissues next to the DA, and the distal lumbar tissue 8 cm away from the lesion (SL), were harvested for single-nucleus suspension.

### Single-nucleus/cell suspension preparation
To isolate nuclei, the protocol was adapted from Sathyamurthy et al. protocols[17]. Tissue samples were homogenized on ice using a glass Dounce tissue grinder (D8938; Sigma) in 1 ml of cold Nuclei EZ lysis buffer (NUC101-1KT; Sigma). Each sample was ground 15 strokes with pestle A, followed by 15 strokes with pestle B. The lysate was then diluted with 3 ml of cold sucrose buffer (0.32 M sucrose) and centrifuged at 3200 g for 10 min. The supernatant was discarded, and the sediment was carefully resuspended with 3 ml of cold sucrose buffer (0.32 M sucrose). Next, the suspension was gently overlaid on 12 ml of density buffer (1 M sucrose) in a 50 ml conical tube and centrifuged at $3200 \times g$ for 20 min. The supernatant was aspirated and discarded, and nuclei deposited on the tube wall were collected with 3 ml of cold PBS. The isolated nuclei were suspended in the Nuclei Suspension Buffer consisting of 1× PBS and 0.01% BSA (Ambion) and 0.1% RNase inhibitor (2313 A, TaKaRa).

For cell preparation, tissues were digested using a papain dissociation system according to the manufacturer's protocol (LS003126, Worthington). Spinal cords were roughly cut into small pieces and incubated in papain solution containing DNase I at 37 °C for 1–1.5 h with occasional pipetting. Albumin-ovomucoid inhibitor solution was used to terminate the enzymatic reaction. The cell suspension was further pipetted to generate a single-cell suspension, which was centrifuged at $400 \times g$ for 5 min to obtain a cell pellet. The cell pellet was then resuspended with 20% Percoll (P1644, Sigma) and centrifuged at $400 \times g$ for 10 min to remove myelin debris. The cells were resuspended in PBS with 0.04% BSA (Ambion) after dissociation.

### Single-nucleus/cell RNA-seq library construction
To assess the quality of the single-cell and single-nucleus suspensions, we measured cell viability and nucleus integrity. Subsequently, cDNA libraries were generated from the suspensions using the Chromium Single Cell V3 Chemistry Library Kit (10X Genomics) following the manufacturer's instructions. Finally, all libraries were sequenced using the Illumina NovaSeq platform to obtain high-quality sequencing data for downstream analyses.

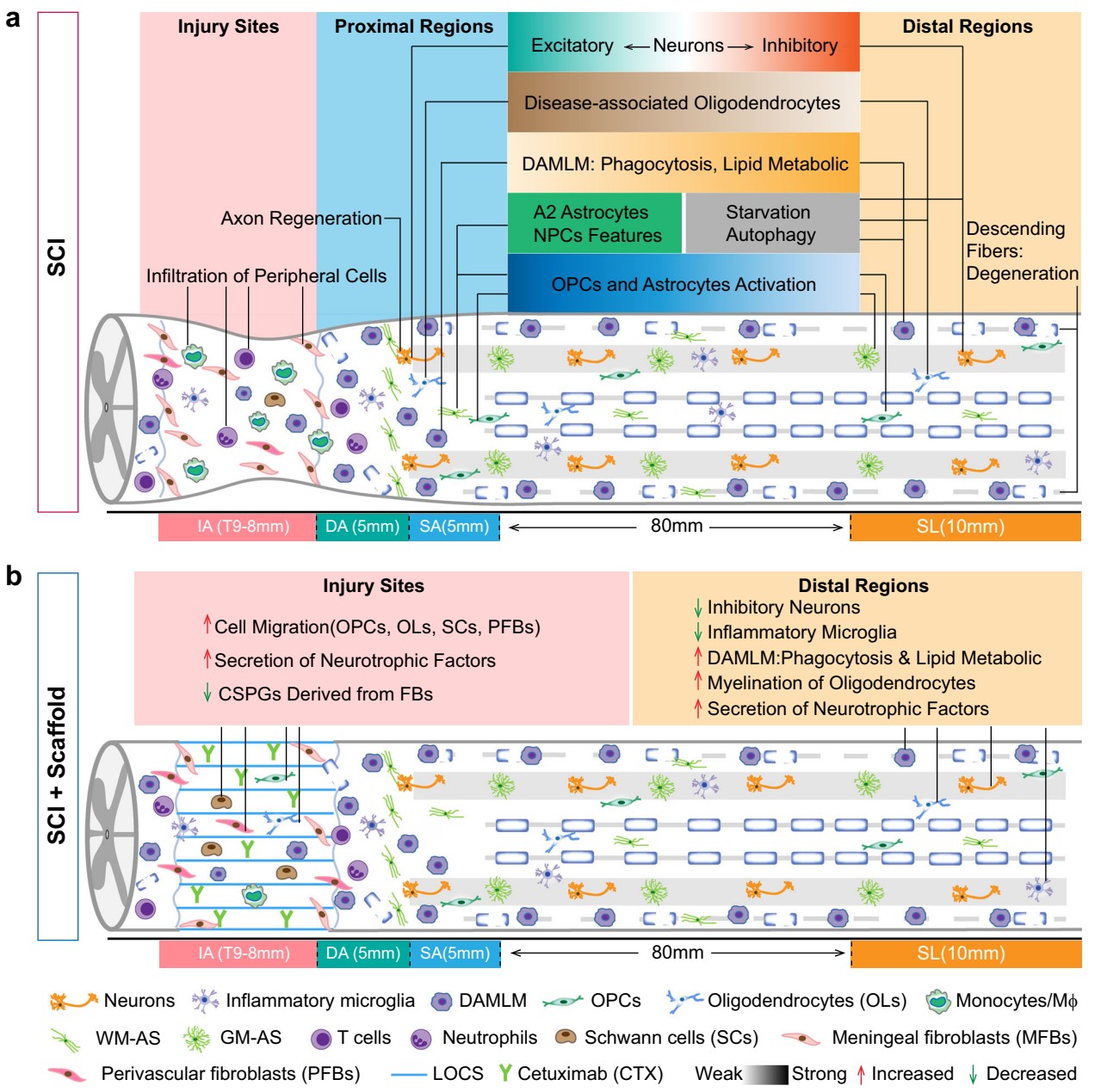

**Fig. 9 | Schematic summarization of cellular responses in the injured rhesus monkey spinal cord without and with functional scaffold implantation. a** Cellular responses in proximal and distal regions of injured rhesus monkey spinal cord. **b** Cellular responses in the injured rhesus monkey spinal cord with functional scaffold implantation. DAMLM disease-associated microglia, Mφ macrophages, WM-AS white matter astrocytes, GM-AS gray matter astrocytes, LOCS linear-ordered collagen scaffold.

## Single-cell/nucleus processing and analysis

To process the sequencing data, we performed barcode filtering and sequence alignment using Cell Ranger[81]. The reads were aligned to the reference genome of *M. mulatta* (Mmul_10). To ensure high-quality data, we removed doublets using DoubletFinder with the default parameters and excluded the 5% of cells most similar to the pseudo-doublets[82]. Next, we loaded the filtered cell-by-gene count matrix into Seurat for downstream analysis. We selected cells that met the following criteria: (1) >500 detected genes; (2) UMI greater than 1000; (3) the percentage of mitochondrial counts <10; (4) the percentage of hemoglobin counts <5; (5) Genes that were expressed in more than 10 cells. The filtered count matrix was loaded into CreateSeuratObject to create a Seurat object, followed by log-normalization of the count matrix using the NormalizeData function. The top 2000 variable genes were identified using the FindVariableFeatures function. Different samples were integrated using the "anchors identify" step and the IntegrateData function to correct batch effects[83]. Unsupervised clustering was performed with the functions FindNeighbors and FindClusters, and clusters were then visualized with t-SNE or UMAP. Differential gene expression analysis among clusters was performed using the Seurat FindAllMarkers or FindMarkers function, with the active assay of the Seurat object set as 'RNA'. Genes with adjusted *P*-values < 0.05 were selected as DEGs. We used several R packages, such as ggplot2, tidyverse, pheatmap, and EnhancedVolcano to assist with the analysis.

## Analysis of specific cell types

To perform further cluster analysis of each specific cell type, we set the active assay of integrated Seurat object as 'integrated' to correct for batch effects. We extracted cell clusters in the integrated Seurat object

using the subset function and repeated the above cluster analysis and DEG analysis steps for each specific cell type.

### Gene Ontology enrichment analysis
GO enrichment analysis was performed with the R package ClusterProfiler[84].

### Pseudo-time analysis
We used monocle version 2.3.6 to estimate lineage differentiation between cell populations using default parameters[85]. DEGs identified by Seurat were used to sort cells in pseudo-time order. We used "DDRTree" to reduce dimensions and "plot_cell_trajectory" to plot the minimum spanning tree.

### Cross-species integration analysis
To compare cellular features between rhesus monkey and human, and mouse, we performed cross-species integration using Seurat. Before using the FindIntegrationAnchors function of Seurat, we reannotated human or mouse genes based on an orthologous genes table between the two species obtained from Biomart. If a human or mouse gene had more than one orthologous gene, we retained only one gene chosen randomly. Finally, Seurat's integration pipeline was conducted on the two species objects with common features.

### Cell−cell interaction analysis
To enable a systematic analysis of cell-cell interactions, we calculated the expression of ligand-receptor pairs in different cell types using CellPhoneDB[86]. Before loading macaque datasets into CellphoneDB, genes were converted into human orthologues by the Ensembl biomaRt package[87]. To decrease the effect of samples with high cell numbers, all samples were randomly down-sampled to the size of the smallest sample. This allowed us to identify potential interactions between different cell types and gain insights into the signaling pathways and regulatory networks involved in cellular processes.

### Immunostaining and lipid droplet staining
The spinal cord was fixed in 4% paraformaldehyde in PBS at 4 °C for 24 h and then dehydrated in successive solutions of 20% and 30% sucrose in PBS for 24 h each. The fixed and dehydrated tissues were embedded in the O.C.T. compound and frozen at −80 °C. Sections were cut using a cryostat microtome (Leica CM3050S, Germany). Cryosections were subjected to antigen retrieval with 50% sodium citrate, followed by blocking with 10% normal donkey serum and 0.1% Triton X-100. Primary antibodies were incubated with sections overnight at 4 °C. The following primary antibodies were used: GFAP (Millipore, MAB360, 1:500), GFAP (Abcam, ab4674, 1:500), NeuN (Abcam, ab177487, 1:500), NeuN (Millipore, MAB377, 1:200), NPY(Abcam, ab221145, 1:500), AIF1 (Wako, 019-19741, 1:500), AIF1 (Abcam, ab5076, 1:500), IL-33 (Abcam, ab207737, 1:200), SLC1A2 (Abcam, ab41621, 1:200), Tuj-1 (Millipore, 05-559, 1:500), GPNMB (Abcam, ab235873, 1:400), VCAN (Invitrogen, MA5-34654, 1:200), GABA (Sigma, A2052, 1:400), NF (Sigma, N4142, 1:400), MBP (Abcam, ab218011, 1:500), APC (Abcam, ab16794, 1:200), SERPINA3 (Abcam, ab180492, 1:400), Ki67 (Abcam, ab15580, 1:500), MPZ (Proteintech, 10572-1-AP, 1:500), SOX9 (Abcam, ab185966, 1:500), and TNF-α (Abcam, ab6671, 1:300). Sections were then washed three times for 5 min with PBS before being incubated in the following secondary antibodies diluted in blocking solution: 488 donkey anti-mouse (Invitrogen, A21202, 1:500), 488 donkey anti-rabbit (Invitrogen, A21206, 1:500), 568 donkey anti-mouse (Invitrogen, A10037, 1:500), 568 donkey anti-rabbit (Invitrogen, A10042, 1:500), 647 donkey anti-chicken (Invitrogen, A78952, 1:500). For lipid droplet staining, sections were washed once in PBS and incubated in PBS with BODIPY 493/503 (1:1000 from a 1 mg ml$^{-1}$ stock solution in dimethylsulfoxide (DMSO), Thermo Fisher) for 15 min. Sections were then washed three times for 5 min with PBS and mounted with Mounting Medium with DAPI (ZSGB-BIO, ZLI-9557). Immunofluorescence images were acquired with a confocal microscope (Leica SP8, Germany).

### TUNEL assay
TUNEL assays were carried out using the Plus Fluorescein In Situ Apoptosis Detection Kit (S7111, Millipore) according to the manufacturer's instructions. Briefly, slices were permeabilized with pre-chilled ethanol: acetic acid (2:1) and subsequently incubated with the TUNEL reaction solution mixture in a humidified 37 °C chamber for 1 h. The reaction was terminated with stop buffer and washed with PBS. Then the slices were incubated with anti-digoxigenin conjugate in a humidified chamber for 30 min at room temperature. Finally, the slices were mounted with Mounting Medium with DAPI (ZSGB-BIO, ZLI-9557). Immunostaining images were acquired with a confocal microscope (Leica SP8, Germany).

### Transmission electron microscopy analysis
Morphological analysis was conducted using transmission electron microscopy (TEM). The lumbar spinal cord tissues fixed with 4% paraformaldehyde were subsequently fixed in 2.5% glutaraldehyde in 0.1 M phosphate buffer (pH 7.3) at 4 °C for 24 h. The embedded tissue was sliced into sections with a thickness of 5 μm thickness, and samples were acquired using a HITACHI H-7650B transmission electron microscope.

### Statistics and reproducibility
For each experiment except scRNA-seq/snRNA-seq, three or more repeats were performed independently. For anatomical and histological work, randomly selected tissue slices from different positions in the rostral-caudal or dorsal-ventral directions were used for statistical analysis. To analyze the data, the cell number, mean fluorescence intensity, and fluorescence area were measured with NIH ImageJ software, with the same threshold set for different samples to ensure consistency. The resulting data points were presented in the graph. GraphPad Prism (GraphPad Software) was used for statistical analyses, with a two-tailed Student's $t$ test used for the comparison between two groups, and one-way ANOVA with Tukey correction for multiple comparisons. $p$-values < 0.05 were considered significant.

### Reporting summary
Further information on research design is available in the Nature Portfolio Reporting Summary linked to this article.

## Data availability
All data associated with this study are present in the paper or the Supplementary Materials. Raw sequencing data and processed expression matrices generated in this study have been deposited to the Gene Expression Omnibus (GEO) under the accession number GSE199669 and GSE228032. The reference genome of M. mulatta (Mmul_10) is available at https://hgdownload.soe.ucsc.edu/downloads.html. The published mouse data is downloaded from Gene Expression Omnibus (GSE172167). Source data are provided with this paper.

## Code availability
There was no custom code development and all software used in the current study are published, open-access, and cited under the relevant method sections. Readers can access the code in the software repositories and documentations via the links provided below: Cell Ranger (https://www.10xgenomics.com/); DoubletFinder (https://github.com/chris-mcginnis-ucsf/DoubletFinder); Seurat (https://satijalab.org/seurat/); Monocle (http://cole-trapnell-lab.github.io/monocle-release/docs/); CellphoneDB (http://www.cellphonedb.org).

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

## Acknowledgements

We are grateful to Y. Shi from CapitalBio Technology for help with scRNA-seq, and J. Zhang from Beijing Institute of Xieerxin Biology Resource for sharing expertize in surgery. This work was supported by the National Natural Science Foundation of China (NSFC) (Grant Nos. 81891002) to J.D. and the Strategic Priority Research Program of the Chinese Academy of Sciences (Grant No. XDA16040700) to J.D. We also thanked the support from the National Natural Science Foundation of China (No. 31970640) to X.W. and Youth Innovation Promotion Association CAS (Grant No. Y202031) to Y.Z. and CAS Project for Young Scientists in Basic Research (Grant No. YSBR073) to Y.Z. and CAMS Innovation Fund for Medical Sciences (Grant No. 2022-I2M-3-002) to J.D.

## Author contributions

J.D., Y.Z., and Z.X. conceived the project and supervised the research. Y.F., S.H., Q.Z., and Z.S. collected the samples. Y.F. and S.H. performed the scRNA-seq experiments. Y.F., X.W., Z.X., Y.Z., and B.C. analyzed the scRNA-seq data. H.Z., X.X., Z.C., and M.Y. carried out the

immunostaining and image data analysis. Y.F., Y.Z., and Z.X. wrote the manuscript. All authors discussed, edited, and proofread the manuscript.

## Competing interests

The authors declare no competing interests.
