## [Peer Review File · Nature Communications]

Single-cell analysis reveals region-heterogeneous responses in rhesus monkey spinal cord with complete injuryREVIEWER COMMENTS

Reviewer #1 (Remarks to the Author):

The manuscript describes a very complex primate study on the effects of scaffold-based treatment on a thoracic spinal complete transection. A single-cell transcriptomics analysis was used to characterize the region-related cellular responses below the injured level. The authors show distinct alterations of a degenerative microenvironment in the distal lumbar tissue, and an improvement by scaffold implantation. Decreasing inhibitory neurons, enhancing phagocytosis, and increasing remyelination are three key factors to remodel the lumbar cord.

Experiments using the described functional collagen scaffold are not especially novel, as a similar experiment was performed and published by the authors in a completely spinal cord transected nonhuman primate model of SCI and chronic SCI patients. Thus, the real novelty of this study is the use of the treatment in primates to explore the molecular changes of multiple cell types. This makes this study possibly more suitable for a neuroscience or a molecular oriented journal.

1. What does Odpi mean? Immediately after the injury or before the injury?
2. The author qualitatively described the region-related cell population dynamic changes in the spinal cord injured monkeys from the acute to the chronic stage. Is there any quantitative analysis to reveal the trends and differences of dynamic changes?
3. The authors state that neurons completely died by 7 days after SCI within 3mm from the IA. However, many NeuN+ signals can be found in this region. It should be clarified.
4. The authors claimed that the downregulation of many myelin-associated genes as well as the newly arisen of oligodendrocyte subtypes impaired myelination ability. It is truly unfortunately that no meticulous examination of myelination capacity is performed. Therefore, it could not reliably confirm the association between the above molecular responses and the degree of myelin damage and resolve the extent to which different cellular changes contribute positively / negatively to myelination capacity.
5. Figure 6a is confusing. It does not appear to demonstrate the spinal cord boundary in SCI group animals. Since it was a complete transection and removal 8mm of the spinal cord, the NF-positive signal in the SCI group in (a) was doubtful. Similarly, in the SCI collagen scaffold group, NF-positive signals only appeared in the unilateral side of the cord. Why does the collagen scaffold implanted in the completely injured cord only exhibit therapeutic effects on one side? The figure is also very descriptive. A quantitative measure and comparison of the lesion sizes, regenerated length, and neurofilament staining fiber density for each group is essential.
6. The lack of correlation between the results of physiological changes and molecular response is a serious defect of this manuscript. Especially in the absence of statistical comparative analysis of the

differences in molecular response, the authors can only speculate that the differences in molecular regulation are the reasons for the alterations in physiological results, but cannot provide sufficient proof to support this hypothesis.

7. In the present work, only two monkeys were used for functional scaffold implantation after SCI. The authors should show the extent of spinal cord regeneration and the degree of physiological changes in each of the treated monkeys. Moreover, the differential effects of functional scaffold implantation on these treated animals and the recovery of the neural fiber bundles should be analyzed on an individual basis. These data are very critical to this kind of nonhuman primate study in which only a limited number of treated subjects can be used. Data points in the figure would be appreciated.

Reviewer #2 (Remarks to the Author):

In this manuscript, the authors performed sc/snRNA-seq to characterize the detailed cellular heterogeneity and dynamics underlying spinal cord lesion in rhesus macaque monkeys. In addition, they revealed a region-dependent heterogeneous response and explained the underlying mechanism of functional recovery after scaffold implantation. Their findings provide comprehensive insights for the development of combined repair strategies to promote proximal tissue regeneration and protect distal spinal cord tissue after SCI. However, there are some issues that need to be addressed before the paper can be considered for publication.

1. The authors should provide a detailed description of scRNA-seq, snRNA-seq and their applications to spinal cord research in the Introduction section. Also, the authors should explain why snRNA-seq was not applied to all samples.
2. On page 5, lines 103-105, the authors show that 11 cell types have been identified, including neurons, astrocytes, oligodendrocytes, OPCs, microglia, monocytes/macrophages, ependymal cells, fibroblasts, endothelial cells, pericytes, and lymphocytes. However, in Fig. 1g there is another cell type, the Schwann cell. The authors should include this cell type in the manuscript and in Fig. 1d & e. Also, the authors should unify the descriptions of lymphocytes, T cells and immune cells. What cell type did “immune” mean in Fig. 1g, T cell, lymphocyte or immune cells (microglia, macrophages, T cells, and neutrophils) as the authors indicated in page 18 line 398-399?
3. Based on gene expression and GO analysis of DEGs, the authors cluster microglia/macrophages into six distinct subtypes, which they call homeostatic microglia, inflammatory microglia, etc. However, the paragraphs in the “Injury induced the continuous activation of microglia into a disease-associated state” should be re-organized to make it more clear and focus on DAM.
4. The authors identified cellular heterogeneity in the injured spinal cord and discussed the function of each cell type subtype. However, how do these cell types or subtypes interact with each other in the lesion microenvironment after injury? The authors should perform additional analysis and discussion.
5. In mice and rats, scRNA-seq and snRNA-seq have been performed to elucidate the microenvironment in the spinal cord after injury. So according to the author’s research in rhesus monkeys, what are the differences between rhesus monkeys and rodents in the lesion microenvironment? The authors should

add a detailed discussion on this point.

6. The entire manuscript should be reorganized and would benefit from editing for syntax and language.

Reviewer #3 (Remarks to the Author):

This manuscript has a strong foundation. It tackles an important topic of how primate (and presumably human) spinal cord reacts to injury and scaffold placement, the central scRNA-seq/snRNA-seq experiment was well designed, and the resulting dataset is a large and powerful resource for the field. However, there are many issues with over-interpreted claims, lack of validation, and a lack of clarity in the data presentation that currently limit the value of this work.

(1) Major claims:

- a. The authors emphasize the effect of “starvation”, “nutritional deficiency”, and “autophagy” in DAMLM and oligodendrocytes but this confounds general markers of cellular stress with a specific cause, for which there is no support.
- b. The authors do not provide convincing evidence for a shift in inhibitory neurons. The slight shifts in percent of inhibitory neurons could easily be random fluctuations and the only supporting evidence is a total fluorescence quantity analysis of GABA in a few fields of view. This type of claim would need to be supported with counts of inhibitory and excitatory neurons in tissue. There is no evidence presented of “neurotransmitter switching”.
- c. I’m confused by the argument that “newly arisen oligodendrocyte subtypes had compromised myelination ability, which may be related to demyelination in SCI”. First, this rests entirely on gene expression rather than a functional description. Second, of course less mature cell types do not express terminal markers of oligodendrocyte and myelin production but this is not because they are compromised. Third, there is no evidence that this shifts with injury.

(2) Specific points:

- a. The authors do not describe the data presentation clearly.
 - i. What is an “enriched GO term ratio”?
 - ii. What is the measure presented in the numerous heatmaps? The legend just says “expression of genes” but why does this include negative values?
 - iii. Are the proportions (of sample?) presented differently for different cell types or is there a scale error in Figure 4d?
 - iv. The logic of which genes to display and emphasize and what their expression may suggest is very confusing. How were the genes chosen for Figure 2g? Why are all values either totally blue or totally red? Does the scale need to be extended to show the dynamic range of the data? How were the genes chosen for Figure 3g? Many of these genes have functions not related to axonogenesis. Was there an unbiased or principled way to select the genes for analysis or is this a curated selection amongst genes that look interesting?
- b. The selection of genes to display should include well established markers for each cell type as a

reference.

c. Although the use of only two animals per condition is completely understandable (and does not need to be changed) this does limit the types of statistical analysis that can be performed. The authors should avoid terms like “significantly” etc.

d. The authors do not present data that the CST area had a lot of “debris” nor do they show proper co-localization of Tuj-1+ signals with microglia. This statement should be removed.

e. The argument about NMOL1/MMOL1 and NMOL2/MMOL3 should be supported with stronger data. The similarity between different populations can be quantified with a correlation matrix or with an unbiased set of oligo markers.

Response to Reviewers:

Thank you very much for your insightful comments regarding our manuscript. We have carefully revised it accordingly, and our responses are provided below on a point-by-point basis. Changes made to the manuscript have been highlighted in the revised version.

Reviewer #1 (Remarks to the Author):

The manuscript describes a very complex primate study on the effects of scaffold-based treatment on a thoracic spinal complete transection. A single-cell transcriptomics analysis was used to characterize the region-related cellular responses below the injured level. The authors show distinct alterations of a degenerative microenvironment in the distal lumbar tissue, and an improvement by scaffold implantation. Decreasing inhibitory neurons, enhancing phagocytosis, and increasing remyelination are three key factors to remodel the lumbar cord.

Experiments using the described functional collagen scaffold are not especially novel, as a similar experiment was performed and published by the authors in a completely spinal cord transected nonhuman primate model of SCI and chronic SCI patients. Thus, the real novelty of this study is the use of the treatment in primates to explore the molecular changes of multiple cell types. This makes this study possibly more suitable for a neuroscience or a molecular oriented journal.

1. What does 0dpi mean? Immediately after the injury or before the injury?

Response:

Thank you for your comments. We apologize for the lack of clarity in our previous manuscript. To clarify, "0dpi" refers to the uninjured tissue prior to injury. We have added a corresponding description in line 95 of the revised manuscript.

2. The author qualitatively described the region-related cell population dynamic changes in the spinal cord injured monkeys from the acute to the chronic stage. Is there any quantitative analysis to reveal the trends and differences of dynamic changes?

Response:

Thank you for your comments. We have made revisions to the manuscript based on your feedback. First, we have added relevant quantitative analysis to the Results section (Lines 106-110). Additionally, we performed immunofluorescence staining and quantitative analysis on tissue sections from various regions (Supplementary Fig. 1) (Lines 111-117), which yielded findings consistent with the RNA-seq data.

Furthermore, we included further quantitative analysis for other results. For instance, immunostaining revealed that dividing microglia (DM) was primarily located in the corticospinal tract (CST) area on the lateral sides of the lumbar spinal cord (33.4%), rather than in the fasciculus gracilis (FG) area on the dorsal column (4.2%) (Supplementary Fig. 2e) (Lines 127-130). We also conducted immunostaining with γ -aminobutyric acid (GABA), an inhibitory neurotransmitter, in the spared lumbar sections, which confirmed an increased proportion of inhibitory interneurons at 6 months after SCI (Fig. 3f) (Lines 207-209).

Moreover, we found that SERPINA3-positive oligodendrocytes were mainly distributed in the CST area and less in the FG area, accounting for approximately 70.9% in the CST area and 23.3% in the FG area at 7dpi among APC-positive oligodendrocytes (Fig. 4f, g) (Lines 258-260). We also conducted an analysis on the distance of axon growth after scaffold implantation (Fig. 6a, b) (Lines 396-397), and these data have been incorporated into the revised manuscript.

3. The authors state that neurons completely died by 7 days after SCI within

3mm from the IA. However, many NeuN+ signals can be found in this region. It should be clarified.

Response:

Thank you for your comments. We have made some changes to the antibody concentration and re-performed tissue immunostaining, as we found that there may have been some non-specific positive signals in our previous images. The revised images are presented in Figure 2c, and they are consistent with the sequencing data that within a 3 mm region away from the IA, almost all neurons died by 7 days after SCI.

4. The authors claimed that the downregulation of many myelin-associated genes as well as the newly arisen of oligodendrocyte subtypes impaired myelination ability. It is truly unfortunately that no meticulous examination of myelination capacity is performed. Therefore, it could not reliably confirm the association between the above molecular responses and the degree of myelin damage and resolve the extent to which different cellular changes contribute positively/negatively to myelination capacity.

Response:

Thank you for your comments. We appreciate your insight that there is not enough evidence to support the idea that different cellular changes positively or negatively contribute to myelination capacity. Therefore, we have reorganized and modified this section of the manuscript (Lines 244 and 253-276). By analyzing the molecular signature of the newly arisen oligodendrocyte subtypes, we found that they were similar to diseased-associated oligodendrocytes (DOLs) previously described in neurodegenerative and autoimmune inflammatory conditions¹. As a result, we have renamed these newly arisen oligodendrocyte subtypes DOL1 and DOL2. While DOL1 and DOL2 expressed lower levels of myelination-related genes compared to MOL1 and MOL3 and were mainly distributed in the CST area rather than the FG area, their myelination ability and

potential positive or negative functions require further investigation. We have also included descriptions of these findings in the Discussion section at Lines 475-485.

5. Figure 6a is confusing. It does not appear to demonstrate the spinal cord boundary in SCI group animals. Since it was a complete transection and removal 8mm of the spinal cord, the NF-positive signal in the SCI group in (a) was doubtful. Similarly, in the SCI collagen scaffold group, NF-positive signals only appeared in the unilateral side of the cord. Why does the collagen scaffold implanted in the completely injured cord only exhibit therapeutic effects on one side? The figure is also very descriptive. A quantitative measure and comparison of the lesion sizes, regenerated length, and neurofilament staining fiber density for each group is essential.

Response:

Thank you for your comments. We performed immunostaining again and used GFAP staining to demonstrate the spinal cord boundary. To address any potential issues with unspecific staining, we optimized the antibody concentration and blocking conditions. Our results showed no obvious NF-positive signals in the SCI group, but we did observe significant axon regeneration in the injury area following scaffold implantation (Fig. 6a).

As for the phenomenon of NF-positive signals appearing only on the unilateral side of the cord, we do not have direct evidence to explain this observation at present. However, we speculate that the uneven contact between the scaffold and the spinal cord may lead to differences in neural regeneration following transplantation.

In response to your suggestion, we have included quantitative analysis comparing the regenerated neurofilament fibers for each group (Fig. 6b). There were no significant differences in lesion size.

6. The lack of correlation between the results of physiological changes and molecular response is a serious defect of this manuscript. Especially in

the absence of statistical comparative analysis of the differences in molecular response, the authors can only speculate that the differences in molecular regulation are the reasons for the alterations in physiological results, but cannot provide sufficient proof to support this hypothesis.

Response:

Thank you for your comments. In this study, we provide comprehensive pathological insights into SCI monkeys with and without scaffold transplantation in spared distal tissues and proximal tissues. Our findings represent an important resource for understanding region-specific pathological characteristics of monkeys after SCI. To establish a correlation between RNA-seq data and physiological changes, we added new experimental data based on immunostaining images and transmission electron microscopy (TEM) images, as well as deep analysis based on statistical and bioinformatics comparison.

Specifically, we conducted the following analyses:

(1) We performed immunofluorescence staining and quantitative analysis on tissue sections from different areas to confirm the dynamic changes observed in RNA-seq data (Supplementary Fig. 1).

(2) We found that disease-associated microglia (DAMLM) were mainly distributed in the descending nerve tract area (CST). Further immunostaining of MKI67 and AIF1 showed that the proportion of dividing microglia was more than 30% in the CST area, while less than 5% in the ascending FG area (Supplementary Fig. 2e).

(3) We observed an increase in the proportion of inhibitory interneurons and oligodendrocytes induced into disease-associated oligodendrocytes (DOLs). The increased proportion of inhibitory interneurons in the spared lumbar was consistent with previous results in mice and cats after SCI²⁻⁴. We further confirmed the gradual increase in the proportion of inhibitory

interneurons by performing immunofluorescence staining with GABA and statistical analysis (Fig. 3f). We performed immunostaining of SERPINA3 with the oligodendrocyte marker APC, and comparative statistical analysis (Fig. 4f, g) showed that DOLs were mainly distributed in the descending nerve tract area (70%), while the proportion in the ascending tract area (FG) was relatively low.

(4) We conducted a comparative analysis between mice and monkeys, which generally shows conserved cellular responses below the lesions of rhesus monkeys and mice after SCI (Supplementary Fig. 9).

(5) Our cell-cell interaction analysis of different regions revealed that intense interactions between scar-forming cells play important roles in pathological changes in proximal regions after spinal cord injury (Supplementary Fig. 10).

(6) After scaffold implantation, we found that oligodendrocytes in the distal lumbar expressed higher levels of myelination genes. Further transmission electron microscopy results showed an improvement in the thickness and integrity of the myelin sheath structure after scaffold implantation (Fig. 8f).

(7) Staining of lipid droplets with BODIPY showed that the accumulation of lipid droplets in microglia of the distal lumbar was reduced after scaffold implantation (Fig. 8j, k).

We hope these data provide support for our current hypothesis. Further validation and deep mechanism investigation will be needed in the future.

7. In the present work, only two monkeys were used for functional scaffold implantation after SCI. The authors should show the extent of spinal cord regeneration and the degree of physiological changes in each of the treated monkeys. Moreover, the differential effects of functional scaffold implantation on these treated animals and the recovery of the neural fiber bundles should be analyzed on an individual basis. These data

are very critical to this kind of nonhuman primate study in which only a limited number of treated subjects can be used. Data points in the figure would be appreciated.

Response:

Thank you for your comments. We apologize for the unclear description in our previous manuscript. In our study, we utilized two monkeys for functional scaffold implantation: one for single-cell sequencing and the other for tissue pathology validation analysis. Additionally, we acknowledged the limited sample size in our discussion (Lines 517-518) and have updated the method section at Lines 527-530 to provide a more detailed description of our animal use. Although our study has the limitation on the sample size of monkeys, the results might be helpful to provide comprehensive insights into pathological changes of the spared distal tissues and proximal tissues after monkey SCI based on the mutual corroboration of sequencing data and immunostaining data. In addition, the cross-species integration analysis between mice and monkeys also supports the conversed cellular responses below the lesion after SCI. We have also taken into account your suggestion and have included data points in all statistical figures.

Reviewer #2 (Remarks to the Author):

In this manuscript, the authors performed sc/snRNA-seq to characterize the detailed cellular heterogeneity and dynamics underlying spinal cord lesion in rhesus macaque monkeys. In addition, they revealed a region-dependent heterogeneous response and explained the underlying mechanism of functional recovery after scaffold implantation. Their findings provide comprehensive insights for the development of combined repair strategies to promote proximal tissue regeneration and protect distal spinal cord tissue after SCI. However, there are some issues that need to be addressed

before the paper can be considered for publication.

1. The authors should provide a detailed description of scRNA-seq, snRNA-seq and their applications to spinal cord research in the Introduction section. Also, the authors should explain why snRNA-seq was not applied to all samples.

Response:

Thank you for your comment. We have updated the Introduction section at Lines 55-62 to include a description of scRNA-seq, snRNA-seq, and their applications in spinal cord research. We apologize for the unclear description in our previous manuscript. Although snRNA-seq has advantages in the unbiased analysis of neural cells in the central nervous system, we found it unsuitable for analyzing IA and DA regions in a complete transection SCI model. The extracted nuclei from these areas included massive nuclei derived from apoptotic cells that did not meet the requirements for library construction and sequencing. As a result, we performed scRNA-seq on IA and DA, which allowed for the analysis of live cells. We have revised the relevant description in the Result section at Lines 90-92.

2. On page 5, lines 103-105, the authors show that 11 cell types have been identified, including neurons, astrocytes, oligodendrocytes, OPCs, microglia, monocytes/macrophages, ependymal cells, fibroblasts, endothelial cells, pericytes, and lymphocytes. However, in Fig. 1g there is another cell type, the Schwann cell. The authors should include this cell type in the manuscript and in Fig. 1d & e. Also, the authors should unify the descriptions of lymphocytes, T cells and immune cells. What cell type did “immune” mean in Fig. 1g, T cell, lymphocyte or immune cells (microglia, macrophages, T cells, and neutrophils) as the authors indicated in page 18 lines 398-399?

Response:

Thank you for your advice. We have added Schwann cells to the manuscript at Line 101 and included them in Fig. 1d-e. We apologize for the confusing descriptions in our previous manuscript and have revised the relevant description in Fig. 1g from T cells to accurately reflect the information presented.

3. Based on gene expression and GO analysis of DEGs, the authors cluster microglia/macrophages into six distinct subtypes, which they call homeostatic microglia, inflammatory microglia, etc. However, the paragraphs in the “Injury induced the continuous activation of microglia into a disease-associated state” should be re-organized to make it more clear and focus on DAM.

Response:

Thank you for your advice. We apologize for the unclear organization in our previous manuscript. In response, we have added a new subheading "DAMLM and Phagocytic Macrophages Exhibited Low Levels of Pro-inflammatory Genes" at Line 157 of the revised manuscript, to better focus on DAM in front of the subheading. The content behind the subheading will focus on the pro-inflammatory M1 or anti-inflammatory M2 phenotype.

4. The authors identified cellular heterogeneity in the injured spinal cord and discussed the function of each cell type subtype. However, how do these cell types or subtypes interact with each other in the lesion microenvironment after injury? The authors should perform additional analysis and discussion.

Response:

Thank you for your advice. We have conducted an analysis of cell-cell interactions in various lesion microenvironments after SCI by adapting CellPhoneDB. This adaptation allowed us to calculate "interaction scores"

based on the average expression levels of ligands and their receptors between two cells. Our cell-cell interaction analysis of different regions revealed intense interactions between scar-forming cells, which play important roles in the pathological changes observed in proximal regions after SCI. In response to your feedback, we have revised relevant descriptions in the Results section at Lines 365-390 (Supplementary Fig. S10). Further investigation of cell interactions will be beneficial for deeper understanding of the pathological mechanisms after spinal cord injury.

5. In mice and rats, scRNA-seq and snRNA-seq have been performed to elucidate the microenvironment in the spinal cord after injury. So according to the author's research in rhesus monkeys, what are the differences between rhesus monkeys and rodents in the lesion microenvironment? The authors should add a detailed discussion on this point.

Response:

Thank you for your advice. We have conducted an integrated analysis of our data with published rodent data in the Results section at Lines 346-364 and Supplementary Fig. S9. Our findings suggest that cells below the lesion in rhesus macaques and mice exhibit overall conserved cellular responses (microglia, neuron, and oligodendrocytes) after SCI. However, we also observed differences in the level of cellular activation and cell subtype quantity between mice and monkeys. These differences may be attributed to species and the different models used (crush or transection). To provide further context on this topic, we have added a detailed discussion in Lines 487-497.

6. The entire manuscript should be reorganized and would benefit from editing for syntax and language.

Response:

Thank you for your comments. We have carefully revised the grammar and language mistakes in the manuscript, and a native speaker has polished the language to improve its quality. The changes have been highlighted in the revised version of the manuscript.

Reviewer #3 (Remarks to the Author):

This manuscript has a strong foundation. It tackles an important topic of how primate (and presumably human) spinal cord reacts to injury and scaffold placement, the central scRNA-seq/snRNA-seq experiment was well designed, and the resulting dataset is a large and powerful resource for the field. However, there are many issues with over-interpreted claims, lack of validation, and a lack of clarity in the data presentation that currently limit the value of this work.

(1) Major claims:

a. The authors emphasize the effect of "starvation", "nutritional deficiency", and "autophagy" in DAMLM and oligodendrocytes but this confounds general markers of cellular stress with a specific cause, for which there is no support.

Response:

Thank you for your comments. The effect of "starvation," "nutritional deficiency," and "autophagy" in the distal lumbar region is the result of GO enrichment analysis of differentially expressed genes. We conducted further transmission electron microscope analysis and observed the presence of autophagosomes and autolysosome-like structures in the distal lumbar segments 6 months after SCI (Fig. 4k). This response is consistent with a cellular stress response when cells experience nutrient deficiency. However, as you mentioned, these results do not provide direct evidence

indicating that these cells are in a state of starvation. As such, we have revised the relevant descriptions to "cellular stress" in the manuscript at Lines 277 and 290-291. We speculated that our complete injury model may have interrupted the blood supply of the longitudinal arteries of the spinal cord, resulting in insufficient nutrition supply to the distal tissues below the injury. Further investigation is needed to validate the hypothesis and explore the molecular mechanism for this phenomenon.

b. The authors do not provide convincing evidence for a shift in inhibitory neurons. The slight shifts in percent of inhibitory neurons could easily be random fluctuations and the only supporting evidence is a total fluorescence quantity analysis of GABA in a few fields of view. This type of claim would need to be supported with counts of inhibitory and excitatory neurons in tissue. There is no evidence presented of "neurotransmitter switching".

Response:

Thank you for your comments. Following your advice, we conducted further immunostaining of GABA in tissues and calculated the proportion of inhibitory interneurons (Fig. 3f). The experimental results were consistent with our data analysis that increased proportion of inhibitory interneurons after SCI in SL. We have added relevant descriptions to the manuscript at Lines 207-209. This result is in line with previous reports in mice and felines that indicate increased inhibitory outputs in the spared lumbar after SCI²⁻⁴. However, as you mentioned, we do not have sufficient evidence to support the "neurotransmitter switching" hypothesis. As a result, we revised the relevant descriptions to "proportion of excitatory and inhibitory interneurons changed" in Lines 200-201.

c. I'm confused by the argument that "newly arisen oligodendrocyte subtypes had compromised myelination ability, which may be related to

demyelination in SCI". First, this rests entirely on gene expression rather than a functional description. Second, of course less mature cell types do not express terminal markers of oligodendrocyte and myelin production but this is not because they are compromised. Third, there is no evidence that this shifts with injury.

Reply:

Thank you for your comments. As you mentioned, we did not have enough evidence to support the hypothesis of newly arisen oligodendrocyte subtypes with compromised myelination ability. Therefore, we have revised this section of the manuscript (Lines 244 and 253-276). By carefully analyzing the molecular signature of the newly arisen oligodendrocyte subtypes, we found that they are similar to diseased-associated oligodendrocytes (DOLs), which have been previously described in neurodegenerative and autoimmune inflammatory conditions¹. We have renamed the newly arisen oligodendrocyte subtypes as diseased-associated oligodendrocytes DOL1 and DOL2. DOL1 and DOL2 expressed lower levels of myelination-related genes compared to MOL1 and MOL3, and they were mainly distributed in the CST area and less in the FG area (Fig. 4f, g). However, the myelination ability and positive or negative functions of DOL1 and DOL2 require further investigation. We have included these descriptions in the Results section at Lines 274-276 and in the Discussion section at Lines 485-486.

(2) Specific points:

- a. The authors do not describe the data presentation clearly.
 - i. What is an "enriched GO term ratio"?

Response:

Thank you for your comments. We apologize for any confusion caused by

the unclear description in the manuscript. The GO ratio represents the number of genes enriched on the GO term divided by the total number of genes that were input for enrichment analysis. To clarify this, we have added relevant descriptions to the figure legend (Supplementary Fig. 2g; Supplementary Fig. 5c; Supplementary Fig. 6d; Supplementary Fig. 7e, f).

ii. What is the measure presented in the numerous heatmaps? The legend just says “expression of genes” but why does this include negative values?

Response:

Thank you for your comments. We apologize for any confusion caused by the unclear description in the manuscript. The heatmaps presented in the figure showed the relative gene expression after a global linear normalization. To improve clarity, we have added relevant descriptions in the figure legend (Fig. 2c, g; Fig. 3h; Fig. 4h, j; Fig. 6h; Fig. 7e; Fig. 8a, c, i; Supplementary Fig. 3c, g, h; Supplementary Fig. 4j; Supplementary Fig. 5d, f; Supplementary Fig. 6a, g; Supplementary Fig. 8c, f; Supplementary Fig. 10j; Supplementary Fig. 11f-h).

iii. Are the proportions (of sample?) presented differently for different cell types or is there a scale error in Figure 4d?

Response:

Thank you for your comments. We apologize for any confusion caused by the unclear description in Figure 4d. The proportion of D-OPC and A-OPC was calculated among all OPCs in each sample, and the proportion of DOL1 and DOL2 was calculated among all oligodendrocytes in each sample. To improve clarity, we have revised the y-axis label in Figure 4d. Additionally, there is no scale error in Figure 4d as our limited sample size.

iv. The logic of which genes to display and emphasize and what their expression may suggest is very confusing. How were the genes chosen for Figure 2g? Why are all values either totally blue or totally red? Does the scale need to be extended to show the dynamic range of the data? How

were the genes chosen for Figure 3g? Many of these genes have functions not related to axonogenesis. Was there an unbiased or principled way to select the genes for analysis or is this a curated selection amongst genes that look interesting?

Response:

Thank you for your comments. The genes displayed in Figure 2g were selected from the corresponding GO-enriched terms in Figure 2f, while the genes displayed in the previous version of Figure 3g (now Figure 3h in the revised manuscript) were selected from the corresponding GO-enriched terms "axonogenesis" in Supplementary Figure 5c. In the previous version of Figure 2g, we used a color scale to represent the relative level of gene expression, where red indicated high expression levels and blue indicated lower expression levels. However, to better illustrate gene expression percentages and average expression levels, we have changed Figure 2g into dot plots and extended the data to show dynamic changes.

b. The selection of genes to display should include well established markers for each cell type as a reference.

Response:

Thank you for your advice. According to your suggestion, we have included MAP2, SYT1, and SNAP25 as neuronal markers in the heatmaps (Fig. 2h and Supplementary Fig. 5d, f) for reference purposes. However, the expression of some established marker genes for oligodendrocytes (MBP), astrocytes (GFAP), and microglia (P2RY12) changed dramatically due to the activation of these cells following SCI. Therefore, we did not include these genes in the heatmaps.

c. Although the use of only two animals per condition is completely understandable (and does not need to be changed) this does limit the types of statistical analysis that can be performed. The authors should avoid terms like "significantly" etc.

Response:

Thank you for your advice. We have revised the manuscript to avoid using words such as "significantly." Additionally, in the discussion (Lines 517-518), we emphasized the limitation of our work due to the small sample size of animals used in this study.

d. The authors do not present data that the CST area had a lot of "debris" nor do they show proper co-localization of Tuj-1+ signals with microglia. This statement should be removed.

Response:

Thank you for your comments. We have replaced the images in Figure 2i with clear, enlarged images. Our results showed that neurofilaments in the ascending nerve tract area (FG) remained intact, while numerous irregularly shaped Tuj-1 positive signals appeared in the descending nerve tract area (CST). Furthermore, we observed co-localization of Tuj-1+ signals with AIF1 signals, which was not observed in uninjured tissue. Following your suggestion, we have removed the word "debris" and modified the sentence to read as follows: " This suggests microglia in the distal lumbar tissue were induced into a phagocytic phenotype surrounding descending axons after SCI." (Lines 155-156)

e. The argument about NMOL1/MMOL1 and NMOL2/MMOL3 should be supported with stronger data. The similarity between different populations can be quantified with a correlation matrix or with an unbiased set of oligo markers.

Response:

Thank you for your advice. Based on your suggestion, we have revised the manuscript for this section (Lines 244 and 253-276). We quantified the similarity between DOLs and MOL2 by conducting a correlation analysis of the gene expression of each subtype, as shown in Supplementary Fig. 7c.

Reference

1. Kenigsbuch, M., *et al.* A shared disease-associated oligodendrocyte signature among multiple CNS pathologies. *Nat Neurosci* **25**, 876-886 (2022).
2. Tillakaratne, N.J.K., *et al.* Increased expression of glutamate decarboxylase (GAD67) in feline lumbar spinal cord after complete thoracic spinal cord transection. *Journal of Neuroscience Research* **60**, 219-230 (2000).
3. Tillakaratne, N.J.K., Leon, R.D.d., Hoang, T.X. & Tobin, A.J. Use-Dependent Modulation of Inhibitory Capacity in the Feline Lumbar Spinal Cord. *The Journal of Neuroscience* **22**, 3130-3143 (2002).
4. Bertels, H., Vicente-Ortiz, G., El Kanbi, K. & Takeoka, A. Neurotransmitter phenotype switching by spinal excitatory interneurons regulates locomotor recovery after spinal cord injury. *Nat Neurosci* **25**, 617-629 (2022).

REVIEWERS' COMMENTS

Reviewer #1 (Remarks to the Author):

The author made revisions based on the review comments, partially addressing the previous issues. However, I still have some concerns:

1. Since many changes are concentrated in CST and less in other regions, has the author explored the reasons for this phenomenon?
2. For regenerated length/NF fiber density, has any obvious difference been observed between the SCI and scaffold implanted groups?
3. What does the “lesion” mean in the Figure 6b? The middle or the caudal boundary of the injured area?
4. Is there any significant difference in Supplementary Fig 1b, 1d, and 2e?

Reviewer #3 (Remarks to the Author):

The authors have addressed my concerns. This work will serve as an important resource for the field.

Response to Reviewers:

Thank you very much for your insightful comments regarding our manuscript. We have carefully revised it accordingly, and our responses are provided below on a point-by-point basis. Changes made to the manuscript have been highlighted in the revised version.

Reviewer #1 (Remarks to the Author):

The author made revisions based on the review comments, partially addressing the previous issues. However, I still have some concerns:

1. Since many changes are concentrated in CST and less in other regions, has the author explored the reasons for this phenomenon?

Response:

Thank you for your comments. The CST, also known as the pyramidal tract, is a collection of descending spinal tracts responsible for transmitting signals from the cerebral cortex to the spinal cord. The cell bodies of the CST are located in the cerebral cortex. Based on this understanding, we can infer that in our complete SCI model, the axons, including those in the CST, are severed from their cell bodies. As a result, the distal portion of these axons below the lesion undergoes stereotypical Wallerian degeneration along with their associated myelin sheaths¹. This degeneration of axons and myelin sheaths triggers the activation of microglia and other cellular responses. It's important to note that the ascending axons with cell bodies located below the lesion will remain intact. Consequently, after SCI, many changes are concentrated in the CST, while fewer changes occur in other regions in the distal lumbar. We have included a relevant discussion on this topic in Lines 478-486 of our study.

2. For regenerated length/NF fiber density, has any obvious difference been observed between the SCI and scaffold implanted groups?

Response:

Thank you for your comments. For regenerated length/NF fiber density, we measure the difference between the two groups by Student's *t*-test at each distance in Figure 6b. The results show obvious differences in NF fiber density at certain distances from the lesion boundary between the two groups and we have revised the Figure 6b to indicate levels of significance with '*'.

3. What does the "lesion" mean in the Figure 6b? The middle or the caudal boundary of the injured area?

Response:

Thank you for your comments. We apologize for the lack of clarity in our previous manuscript. The x-axis label of Figure 6b describes the distance (mm) of NF⁺ axon growth past the rostral boundary of the injured area. We have revised it in Figure 6b and the corresponding figure legend.

4. Is there any significant difference in Supplementary Fig 1b, 1d, and 2e?

Response:

Thank you for your comments. We measure the difference of cellular proportion and number among different time points after SCI by One-Way ANOVA in Supplementary Fig 1b, 1d and measure the difference of dividing microglia proportion between FG and CST area by Student's *t*-test in Supplementary Fig 2e. We have revised Supplementary Fig 1b, 1d, and 2e to indicate levels of significance with '*' or '#'.

Reviewer #3 (Remarks to the Author):

The authors have addressed my concerns. This work will serve as an important resource for the field.

Response:

Thank you very much for your insightful comments.

Reference:

1. Tran, A.P., Warren, P.M. & Silver, J. The Biology of Regeneration Failure and Success After Spinal Cord Injury. *Physiol Rev* **98**, 881-917 (2018).